# The evolution of a counter-defense mechanism in a virus constrains its host range

Sriram Srikant[1†], Chantal K Guegler[1†], Michael T Laub[1,2]*

[1]Department of Biology, Massachusetts Institute of Technology, Cambridge, United States; [2]Department of Biology, Howard Hughes Medical Institute, Massachusetts Institute of Technology, Cambridge, United States

**Abstract** Bacteria use diverse immunity mechanisms to defend themselves against their viral predators, bacteriophages. In turn, phages can acquire counter-defense systems, but it remains unclear how such mechanisms arise and what factors constrain viral evolution. Here, we experimentally evolved T4 phage to overcome a phage-defensive toxin-antitoxin system, *toxIN*, in *Escherichia coli*. Through recombination, T4 rapidly acquires segmental amplifications of a previously uncharacterized gene, now named *tifA*, encoding an inhibitor of the toxin, ToxN. These amplifications subsequently drive large deletions elsewhere in T4's genome to maintain a genome size compatible with capsid packaging. The deleted regions include accessory genes that help T4 overcome defense systems in alternative hosts. Thus, our results reveal a trade-off in viral evolution; the emergence of one counter-defense mechanism can lead to loss of other such mechanisms, thereby constraining host range. We propose that the accessory genomes of viruses reflect the integrated evolutionary history of the hosts they infected.

## Editor's evaluation

This manuscript will be of interest to researchers in the phage-microbial host interaction field. Notably, the interplay between bacteria and their viral predators has regained broad interest in recent years given the discovery of numerous innate immunity-like phage defense systems. The identification of phage-mediated counter-defense strategies is therefore not only of prime importance for our basic understanding of predator-prey arms races but also for medical applications such as phage therapy.

**\*For correspondence:**
laub@mit.edu

[†]These authors contributed equally to this work

## Introduction

Bacteria face the frequent threat of phage predation and have consequently evolved a diverse arsenal of anti-phage defense systems (*Bernheim and Sorek, 2020*; *Tal and Sorek, 2022*; *Wein and Sorek, 2022*). These defense systems have, in turn, driven the selection of counter-defense mechanisms in phage, underscoring the intense arms race between phages and their bacterial hosts (*Samson et al., 2013*; *Hampton et al., 2020*). There are several well-characterized examples of anti-restriction modification (RM) and anti-CRISPR proteins (*Stanley and Maxwell, 2018*). However, there is an ever-growing inventory of phage defense mechanisms in bacteria and how phages counteract these diverse systems is poorly understood (*Samson et al., 2013*). More generally, it remains unclear how phages can rapidly acquire resistance when confronted with a new defense system and what types of mutations and mechanisms are responsible. Probing how phage overcome bacterial defense systems

will help reveal the molecular basis of bacteria–phage coevolution and may inform efforts to develop phage therapies for treating antibiotic-resistant bacterial infections.

An emerging class of potent phage defense genes are toxin–antitoxin (TA) systems, which typically comprise a protein toxin that is directly neutralized by its cognate antitoxin (*Harms et al., 2018*). For defensive TA systems, phage infection activates or triggers the release of the toxin, which can then block phage development (*Guegler and Laub, 2021*). Although the initially infected cell typically does not survive, particularly if the toxin also inhibits host cell processes during infection, these defensive TA systems prevent the release of new, mature virions, thereby preventing spread of an infection through a population.

How can a phage evolve to overcome a defensive TA system? There are three general possibilities: (1) a phage prevents activation of the TA system, (2) a phage becomes resistant to the action of the toxin, or (3) a phage acquires or modifies a factor that can inhibit or degrade the toxin. As TA systems may be activated by and target core phage processes, the latter is a potentially powerful mechanism because it does not require mutations in essential phage genes. Notably, like eukaryotic viruses, phages often harbor many 'accessory genes' that are not formally required for phage propagation in a naïve host in permissive conditions. For instance, T4 encodes nearly 300 proteins, only 62 of which are required to produce a new virion in common laboratory conditions (*Miller et al., 2003*). The remaining genes are mostly of unknown function, but their presence in many related T4-like phages suggests they may play key roles in host specificity by overcoming host defense systems or enabling replication in specific growth conditions.

There are several examples of phage-encoded anti-TA factors. The T4 gene *dmd* encodes a direct inhibitor of the RnlA toxin from the *Escherichia coli* K12 type II TA system RnlAB (*Otsuka and Yonesaki, 2012*; *Wan et al., 2016*). Consequently, RnlAB only protects *E. coli* against T4 phage lacking *dmd* (gene *61.5*). More recently, T-even phages including T4 were shown to encode an inhibitor (AdfA, encoded by gene *61.2* in T4) of DarT toxins (*LeRoux et al., 2022*). In *Pectobacterium atrosepticum*, the type III TA system *toxIN,* involving an endoribonuclease toxin and RNA antitoxin, protects against infection by several phages, including ΦTE and ΦA2. Selection for ΦTE mutants that overcome the *P. atrosepticum toxIN* system identified phages that either expressed additional copies of a locus resembling the *toxI* antitoxin or that had acquired *toxI* from the *toxIN* system directly (*Blower et al., 2012*). This study indicates that phages may be poised to readily evolve mechanisms to overcome defensive TA systems.

Here, we evolved T4 phage to overcome a *toxIN* homolog from a natural plasmid in an environmental *E. coli* isolate that can provide *E. coli* MG1655 with potent defense (*Fineran et al., 2009*; *Guegler and Laub, 2021*). Using an experimental evolution approach adapted from that used in early phage therapy initiatives (*Appelmans, 1921*; *Burrowes et al., 2019*), we find that T4 can rapidly overcome *toxIN* through the segmental amplification and increased expression of *tifA* (previously *61.4*), which encodes a protein inhibitor of ToxN. Strikingly, these segmental amplifications led to T4 genome instability, likely because an increased genome size is incompatible with the fixed capsid size of T4. The evolved phages subsequently acquired compensatory deletions that restored genome size, with the set of deleted genes varying among replicate evolved populations. These deletions compromise the ability of T4 to infect other strains of *E. coli,* often because the lost genes encode factors that overcome other, strain-specific defense systems. In sum, our work reveals the genome dynamics underlying the emergence of an anti-defense mechanism in T4 and demonstrates an evolutionary trade-off in which selection to infect one host can compromise infection of others.

## Results

### T4 can evolve to overcome *toxIN*-mediated defense in *E. coli*

We previously identified and characterized *toxIN*$_{Ec}$ (hereafter *toxIN* for simplicity), a type III TA system from an environmental isolate of *E. coli* that strongly protects *E. coli* MG1655 against T4 infection (*Figure 1A*; *Guegler and Laub, 2021*). To evolve T4 to overcome *toxIN*, we adapted the Appelmans protocol originally developed in phage therapy to evolve phage to replicate on resistant pathogenic hosts (*Appelmans, 1921*; *Burrowes et al., 2019*; *Mapes et al., 2016*). Briefly, our approach involved replicating T4 on both a sensitive host (-*toxIN*), which maintains the phage population size and generates diversity, and on a resistant host (+*toxIN*) to select for phages that can overcome the defense

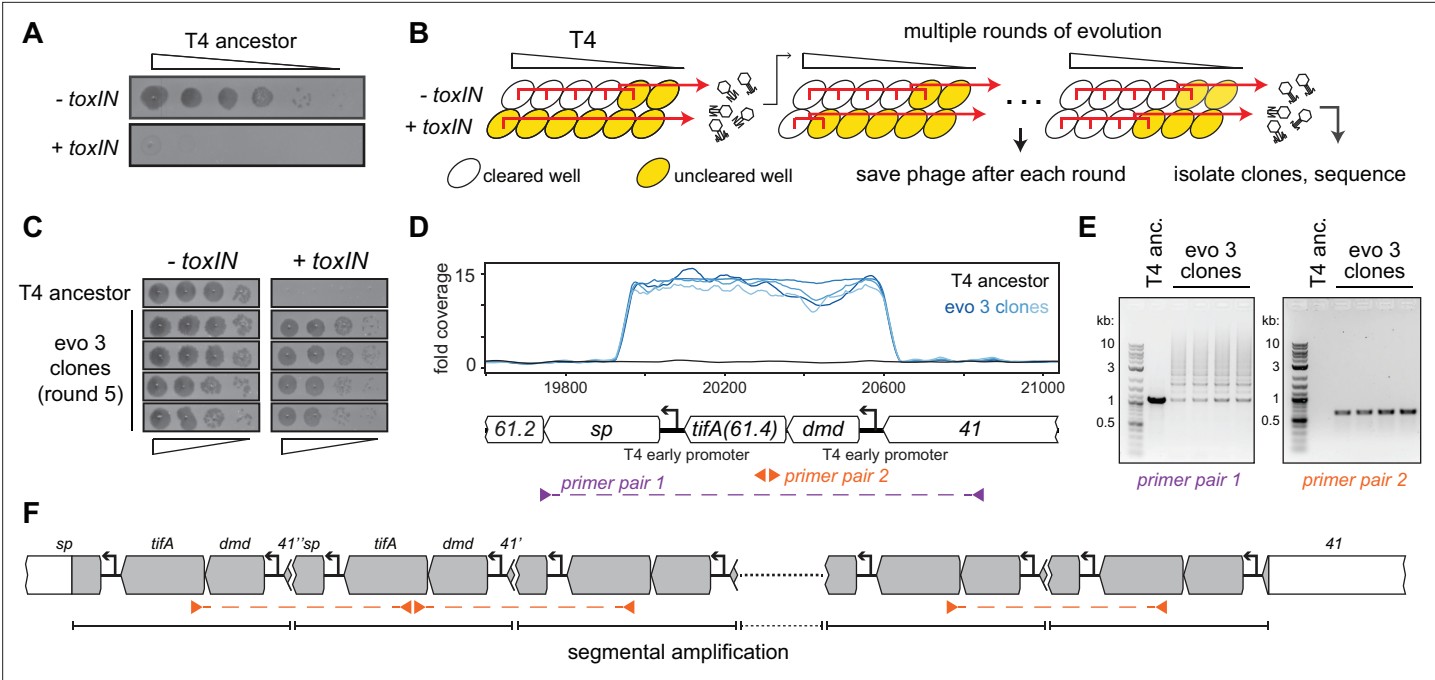

**Figure 1.** T4 rapidly evolves to overcome *toxIN* by segmental amplification of the *dmd-tifA(61.4)* locus. (**A**) T4 is restricted by +*toxIN E. coli*. Serial dilution plaquing assay of T4 spotted on -*toxIN* and +*toxIN* cells. (**B**) Phage evolution protocol to select for T4 that overcomes *toxIN*-mediated defense. Serial dilutions of T4 are used to infect +*toxIN* and -*toxIN* cells at multiple multiplicities of infection (MOIs). The ancestral T4 only clears wells containing -*toxIN* cells. Phage from all cleared wells, and the last uncleared well, are pooled and used to inoculate the next round of evolution. As the population evolves, it can also infect and clear +*toxIN* cells. Phage populations from each round are saved to maintain a fossil record; individual clones are isolated on +*toxIN* cells and sequenced to identify mutations, leading to *toxIN* resistance. (**C**) Comparison of plaquing efficiency for ancestral T4 and T4 isolates from one of the evolved populations after five rounds on -*toxIN* and +*toxIN* cells. (**D**) Read coverage at the *dmd-tifA(61.4)* locus following genome sequencing of multiple clones from one of the evolved populations. Primer pairs used in panel (**E**) to interrogate the segmental amplification are shown below the locus map. (**E**) PCR products using the primer pairs indicated in panel (**D**) for ancestral T4 and evolved clones. (**F**) Schematic of segmental amplification of *dmd-tifA* locus in evo 3 clones highlighting binding of primer pair 2 producing a cross-repeat product.

The online version of this article includes the following source data and figure supplement(s) for figure 1:

**Source data 1.** Uncropped and labeled gel for *Figure 1E*, left (primer pair 1).

**Source data 2.** Uncropped and labeled gel for *Figure 1E*, right (primer pair 2).

**Figure supplement 1.** Experimental evolution of T4 to overcome *toxIN*.

**Figure supplement 2.** Segmental amplifications and deletions arising in evolved T4 clones.

system (*Figure 1B*). During each round of evolution, serial dilutions of the phage population (six serial tenfold dilutions producing a range of $10^7$–$10^1$ phage) were each inoculated with $10^5$ host cells to produce a range of multiplicities of infection (MOIs, defined as the ratio of phages to bacteria) from $10^2$ to $10^{-4}$ in 96-well plates and then grown for 16–20 hr. This protocol allows phage to evolve by a combination of point mutations and recombination events within a phage genome, as well as recombination between co-infecting phage. The clearing of cultures across MOIs provided a visual readout of the phage population evolving to infect the resistant host (*Figure 1B*, *Figure 1—figure supplement 1A*). Throughout the evolution protocol, the T4 population titer remained at $10^5$–$10^6$ pfu/µL, with an estimated 1–3 phage infection generations occurring in wells from high to low MOI within a single round of evolution (see 'Materials and methods'). We set up five replicate evolutions of T4 to overcome *toxIN* carried on a medium-copy plasmid in *E. coli* MG1655 (+*toxIN*), and a control population replicating only on -*toxIN* cells (MG1655 with an empty vector) (*Figure 1—figure supplement 1A*). We plated serial dilutions of the phage populations after each round on lawns of +*toxIN* cells to identify *toxIN*-resistant phage clones as they arose and fixed (*Figure 1—figure supplement 1B*).

For one of the populations (evo 3), phage that could form plaques on lawns of +*toxIN* cells arose after five rounds of evolution, an estimated 5–15 infection generations. From this population, we isolated phages from four individual plaques on +*toxIN* lawns and confirmed that these evolved

phages plaque as efficiently on +*toxIN* cells as on the control strain -*toxIN* (*Figure 1C*). To identify the mutations responsible for overcoming *toxIN*, we fully sequenced the genomes of these four evolved clones. No individual gene was mutated in all of the evolved clones. However, our sequencing revealed increased read coverage of a two-gene locus in the genome of each evolved clone (*Figure 1D*). This locus consists of two genes, *dmd (61.5)* and *61.4*, driven by an early T4 promoter (*Liebig and Rüger*, *1989*; *Miller et al., 2003*). As noted above, Dmd is an inhibitor of the RnlA toxin of the RnlAB TA system (*Otsuka and Yonesaki, 2012*) and *61.4* encodes a putative 85 amino acid protein of unknown function, which we have renamed *tifA* (*ToxN inhibitory factor A*) based on the studies below.

The increased read coverage of the *dmd-tifA* locus could indicate the presence of multiple copies of this locus scattered throughout the T4 genome or a local, segmental amplification of this locus. PCR using primers flanking this locus produced a ladder of products, indicating a segmental amplification generating tandem repeats (*Figure 1E,F*, *Figure 1—figure supplement 2B*). PCR using divergent primers in the middle of the amplified region generated a band for the evolved clones but not the T4 ancestor (*Figure 1E*). This band is produced by primers annealing to neighboring repeats and thus reflects the size of a single repeat in the amplification.

## TifA is a protein inhibitor of ToxN

We hypothesized that the segmental amplifications in our T4 escape mutants led to increased expression of either *dmd* or *tifA*, allowing the phage to overcome *toxIN*. To determine which gene was

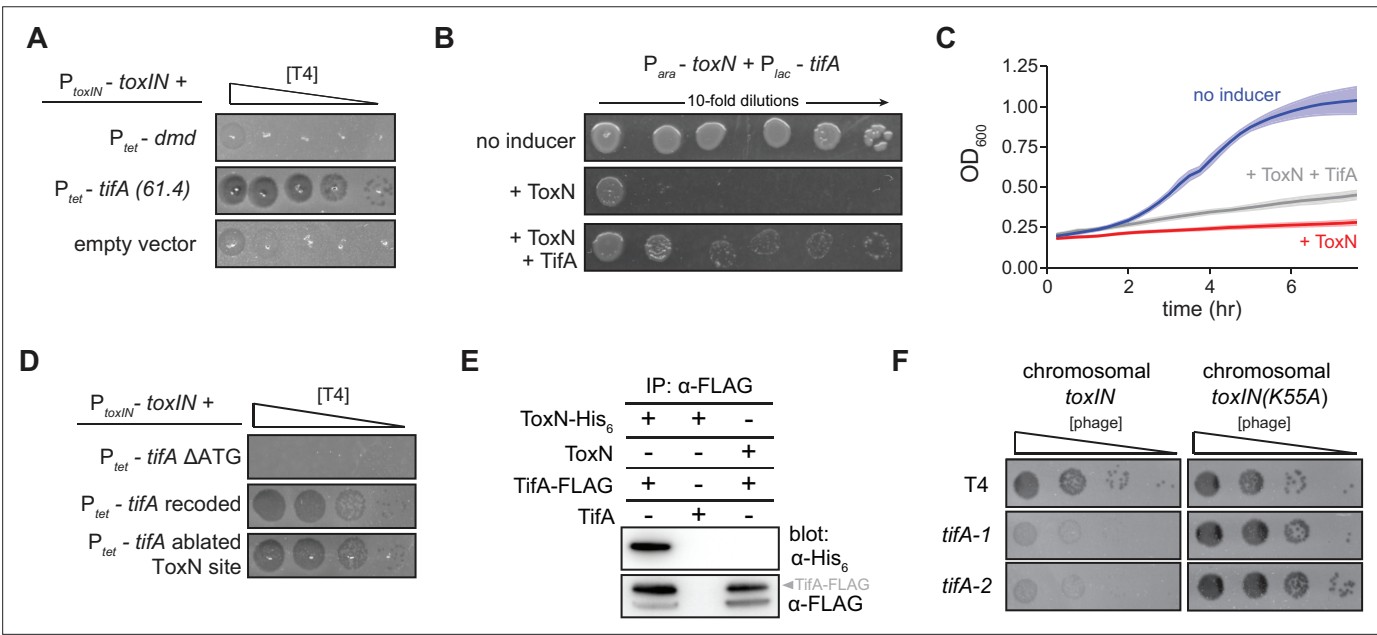

**Figure 2.** TifA (gp61.4) inhibits ToxN toxicity. (**A**) Serial dilution plaquing assay of T4 ancestor spotted on +*toxIN* cells also harboring a plasmid containing *dmd* or *tifA(61.4)*, or an empty plasmid. Anhydrotetracycline was added to the plates to induce *dmd* or *tifA*. (**B, C**) Representative plating assay (**B**) and growth curves (**C**) showing TifA rescue of ToxN. Plasmids harboring *toxN* and *tifA* under arabinose- and IPTG-inducible promoters, respectively, were transformed into *E. coli* MG1655. ToxN, ToxN, and TifA, or neither were induced as indicated. For growth curves in (**C**), data are the average of three technical replicates each of four biological replicates, with shaded areas indicating standard deviation. (**D**) Serial dilution plaquing assay of T4 ancestor spotted on +*toxIN* cells containing plasmids expressing the *tifA* variants indicated. (**E**) Western blot of ToxN-His$_6$ and TifA-FLAG (or untagged controls) following an anti-FLAG coimmunoprecipitation of T4-infected cells co-producing ToxN and TifA. The arrowhead highlights the band with a molecular weight matching that expected for TifA-FLAG. The identity of the band below is unknown but may represent a processed or truncated portion of TifA-FLAG. (**F**) Serial dilution plaquing assay of T4 ancestor and two *tifA* mutants on *E. coli* MG1655 with chromosomally encoded *toxIN* or *toxIN(K55A)*, which harbors an active-site mutation in ToxN.

The online version of this article includes the following source data and figure supplement(s) for figure 2:

**Source data 1.** Uncropped and labeled chemiluminescent Western blot for *Figure 2E*, top (anti-His$_6$ blot).

**Source data 2.** Uncropped and labeled chemiluminescent Western blot (low and high contrast) for *Figure 2E*, bottom (anti-FLAG blot).

**Figure supplement 1.** ToxN-His$_6$ and TifA-FLAG are functional in the context of T4 infection.

**Figure supplement 2.** Generation of T4 clones with mutations in *tifA*.

responsible, we cloned *dmd* and *tifA* separately into a vector containing an inducible promoter and transformed each plasmid into +*toxIN* cells. We then infected both strains with the ancestral, wild-type T4 in the presence of inducer and compared their efficiency of plaquing (EOP) (*Figure 2A*). The ability of *toxIN* to inhibit T4 infection was completely lost following induction of *tifA*, suggesting that this gene was sufficient to prevent ToxN-mediated defense. Overexpressing *dmd* did not allow T4 to replicate in the presence of *toxIN*, indicating that this gene did not contribute to escape, but was likely included in the segmental amplification as the promoter for *tifA* lies upstream of *dmd* (*Figure 1D*).

Our results suggested that *tifA* is a phage-encoded factor that, upon overexpression, can offset ToxN activity during T4 infection. To test whether *tifA* encodes an antitoxin for ToxN, we cloned *toxN* into a vector that allowed for its inducible expression and co-transformed this plasmid into *E. coli* MG1655 with a plasmid containing *tifA* controlled by a separate, inducible promoter. As expected for a bacteriostatic toxin, inducing ToxN in uninfected cells inhibited the growth of *E. coli*. Co-induction of *tifA* was sufficient to partially rescue this toxicity, on both solid media and in culture (*Figure 2B,C*), indicating that *tifA* is a T4-encoded antitoxin for ToxN.

Given that the native antitoxin *toxI* is an untranslated RNA that binds and neutralizes ToxN, we sought to determine whether *tifA* encodes a protein or an RNA antitoxin of ToxN (*Figure 2D*). Expressing a version of *tifA* in which the start codon was deleted (*tifA ΔATG*) could no longer restore T4 infection of cells harboring *toxIN,* suggesting that *tifA* must be translated to function. Additionally, we found that a recoded variant of *tifA* that has a different mRNA sequence (55 of 258 nucleotides changed) but produces the same protein sequence could still restore T4 infection of +*toxIN* cells, indicating that the protein product of *tifA* was sufficient for ToxN inhibition. Finally, to verify that the *tifA* mRNA does not act as a competitive substrate for inhibition of ToxN, an endoribonuclease, we ablated a putative ToxN cleavage site at the 3′ end of *tifA*. Expression of this variant did not interfere with inhibition of ToxN, confirming that it is the protein product, and not the mRNA, of *tifA* that counters *toxIN* in our evolved T4 clones. It is these results that prompted us to rename gene *61.4* to *tifA*.

## TifA interacts with ToxN *in vivo*

To test whether TifA interacts directly with ToxN, we generated a strain producing ToxN from the *toxIN* locus with a C-terminal His$_6$ tag and TifA with a C-terminal FLAG tag. We first verified that these tags did not affect activity of either protein (*Figure 2—figure supplement 1*). We then infected cells with T4, lysed the cells, and performed an immunoprecipitation with anti-FLAG antibodies to isolate TifA and any interacting proteins. Western blotting with an anti-FLAG antibody confirmed that TifA was present, as expected, and blotting with an anti-His$_6$ antibody demonstrated that ToxN was also recovered (*Figure 2E*). These results suggest that TifA interacts with ToxN, either directly or indirectly, *in vivo*.

If T4 encodes an inhibitor of ToxN, why is *toxIN* normally able to defend against wild-type T4? We reasoned that, because *toxIN* is encoded on a medium-copy (~20/cell) plasmid in our system, the single, native copy of *tifA* in T4 might not be sufficient to overcome ToxN. To test this idea, we introduced either *toxIN* or *toxIN(K55A),* which produces an enzymatically dead ToxN, into the *E. coli* MG1655 chromosome and infected both strains with T4 (*Figure 2F*). In this case, *toxIN* was no longer sufficient to protect against wild-type T4 infection. To confirm that this infection requires *tifA*, we used a CRISPR-Cas9 system to generate two mutant strains of T4 (see 'Materials and methods') containing either a 98 bp deletion or 5 bp insertion disrupting the *tifA* open-reading frame (*Figure 2—figure supplement 2*). Both T4 mutant strains exhibited an ~100-fold decrease in plaquing efficiency on a strain containing chromosomal *toxIN*, but were comparable to the wild-type T4 when tested on a *toxIN(K55A)* control strain (*Figure 2F*). These results suggest that native *tifA* expression is sufficient to overcome ToxN produced from the chromosome and that the segmental amplification that arose during our evolution experiments reflects a pressure to overcome the stronger expression of plasmid-borne *toxIN*.

## TifA homologs from other T4-like coliphage inhibit ToxN

Homologs of *tifA* are found adjacent to *dmd* homologs in the genomes of the coliphage T2, T6, and RB69, all of which are closely related to T4 (*Figure 3A*), and in other types of phages that infect a diverse set of bacterial hosts (*Figure 3—figure supplement 1*). TifA homologs share a common

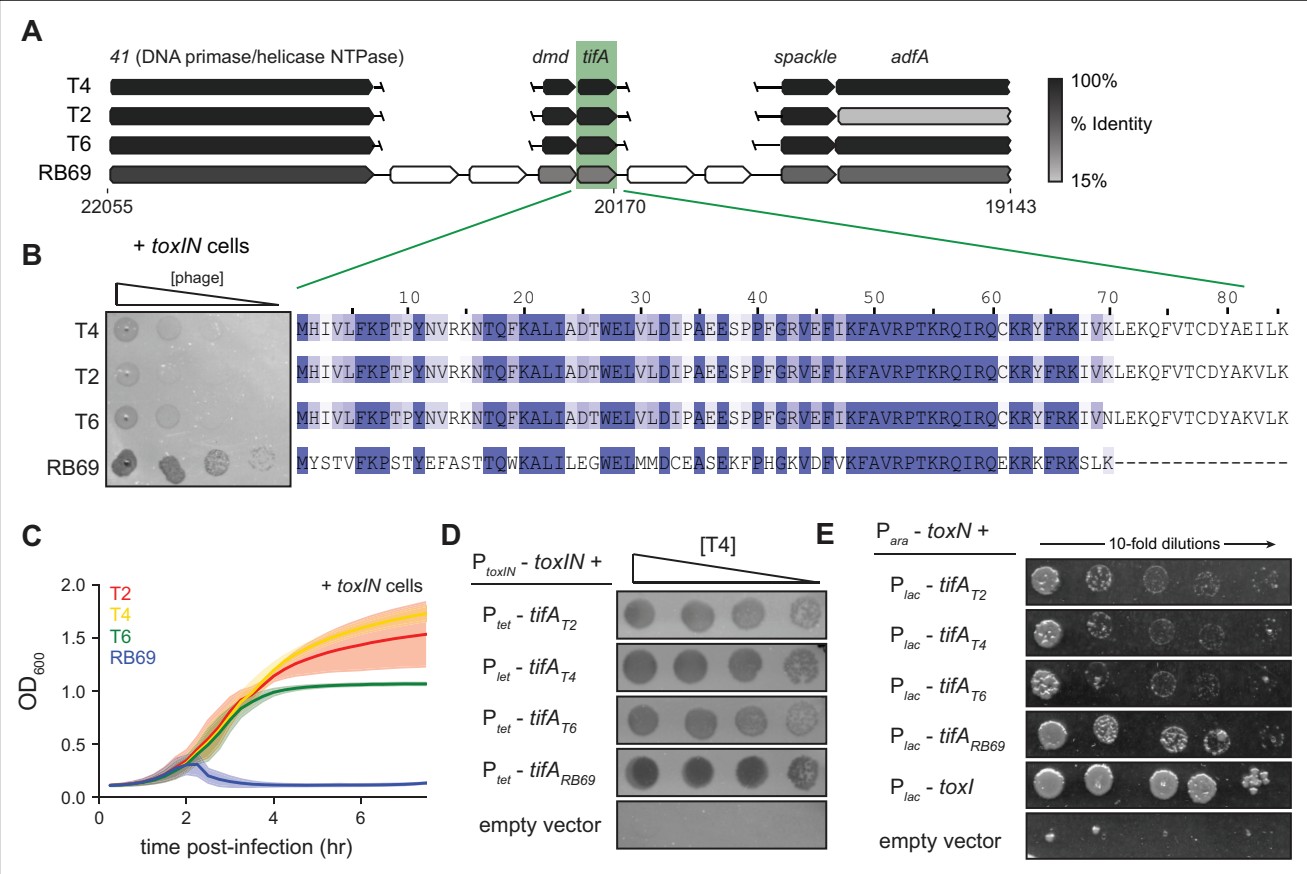

**Figure 3.** RB69 TifA is a more potent inhibitor of ToxN than the T4 homolog. (**A**) Schematic of syntenic region containing *tifA* in closely related T-even phages. Homologous genes are aligned vertically and colored by % identity to the homolog in T4. (**B**) (Left) Serial dilution plaquing assays of T4, T2, T6, RB69 on *+toxIN* cells. (Right) Multiple-sequence alignment of TifA homologs from T4, T2, T6, and RB69. (**C**) Growth curves following infection of *+toxIN* cells with T4, T2, T6, or RB69 phage, each at a multiplicity of infection (MOI) of $10^{-3}$. Data are the average of three biological replicates, with shaded areas indicating standard deviation. (**D**) Serial dilutions of T4 ancestor spotted on *+toxIN* cells expressing *tifA* homologs from T2, T4, T6, RB69, or harboring an empty vector. (**E**) Serial dilutions of *E. coli* cells expressing *toxN* and the *tifA* homolog from T2, T4, T6, or RB69, *toxI*, or alone.

The online version of this article includes the following figure supplement(s) for figure 3:

**Figure supplement 1.** Homologs of *tifA* are found in other phage genomes.

**Figure supplement 2.** TifA homologs from T-even phages inhibit ToxN, with TifA from RB69 providing the strongest inhibition.

architecture, with a central region containing a cluster of positively charged residues (*Figure 3B*, *Figure 3—figure supplement 1*). Although the T2, T4, and T6 homologs are nearly identical to one another, RB69 TifA is more divergent (56% identity) and lacks the C-terminal extension shared by the other three (*Figure 3B*). Interestingly, plasmid-borne *toxIN* strongly protects *E. coli* against T2, T4, and T6 infection, but offers substantially less protection against RB69, as measured by plaquing efficiency and cell death in shaking cultures (*Figure 3B, C*, *Figure 3—figure supplement 2A, B*). Thus, we hypothesized that the more divergent TifA from RB69 may be a more potent inhibitor of ToxN.

To test this hypothesis, we cloned each *tifA* homolog from T2, T4, T6, and RB69 into a plasmid with an inducible promoter and transformed those plasmids into *+toxIN* cells. We then compared the ability of wild-type T4 to infect each strain (*Figure 3C*). Overexpressing any of the four TifA variants allowed T4 to form plaques on *+toxIN* lawns (*Figure 3D*), with cells producing RB69 TifA producing the clearest plaques. We also tested each variant for protection against T2 infection, again finding that RB69 TifA produced the clearest plaques and led to the highest EOP on *+toxIN* lawns (*Figure 3—figure supplement 2C*). Finally, we expressed each *tifA* homolog in uninfected cells ectopically producing ToxN and found that all the TifA homologs rescued cells from ToxN-mediated toxicity to some extent (*Figure 3E*). We conclude that the TifA homolog from each of these T-even phages is

a ToxN inhibitor, but that RB69 may encode a more potent inhibitor, enabling RB69 phage to be least affected by the *toxIN* system we tested.

## T4 genome evolution proceeds through frequent recombination events

For the phages from population 3 that were initially isolated and sequenced after five rounds of evolution, we noticed that individual plaques exhibited substantial variability in size (*Figure 1—figure supplement 2A*), suggestive of some genomic variation during isolation. We therefore

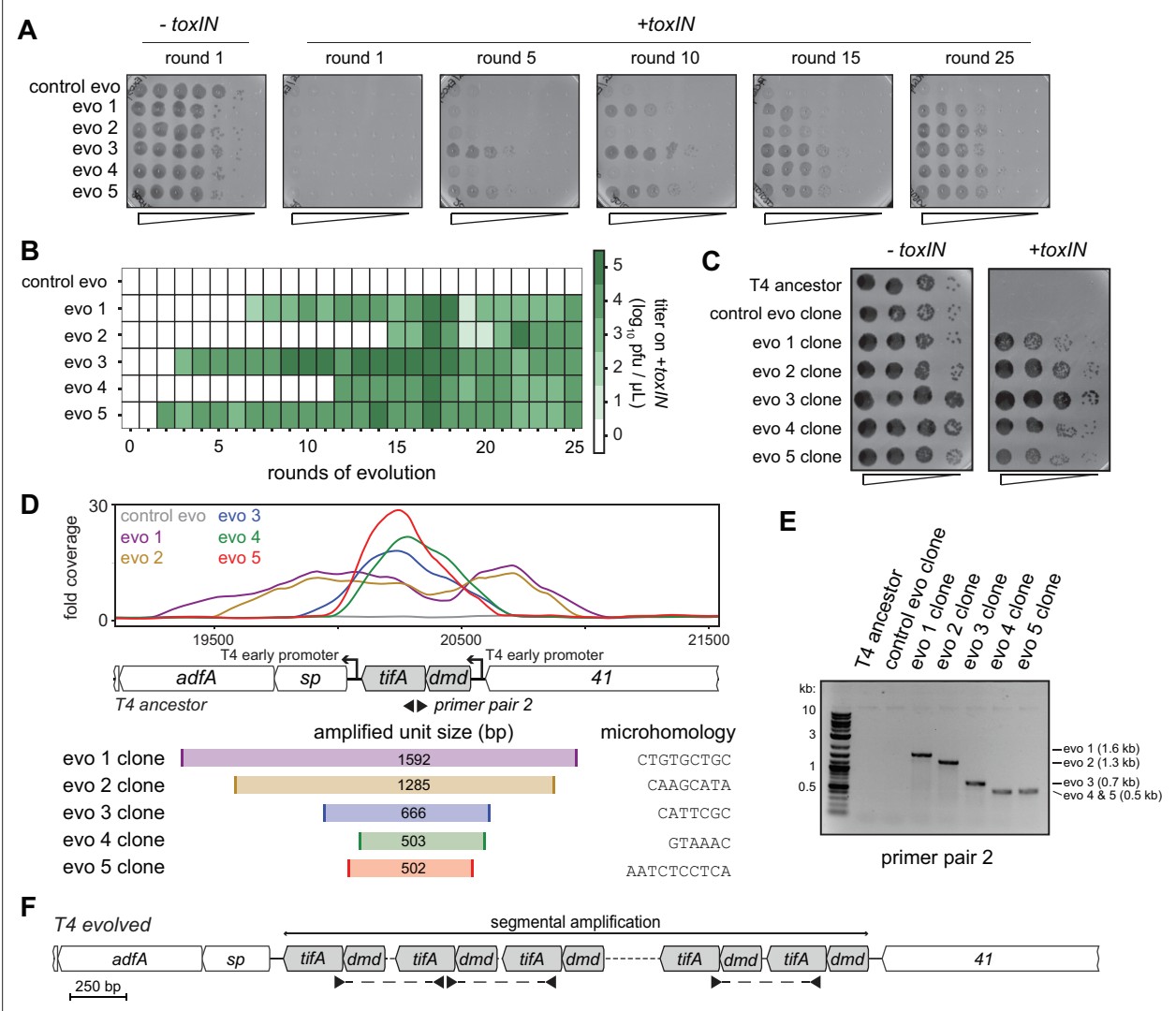

**Figure 4.** Unique recombination events generate *dmd-tifA* amplifications in replicate T4 evolution experiments. (**A**) Serial dilutions of evolving populations (one control evolution and five replicate populations) after rounds 1, 5, 10, 15, and 25 spotted on +*toxIN* cells. (**B**) Heat map summarizing the titer of *toxIN*-resistant T4 as they arise in the independent evolving populations. Titer was estimated by serial dilutions of evolving populations spotted on +*toxIN* cells (as in panel **A**). (**C**) Serial dilutions of isolated clones from evolved populations spotted on -*toxIN* and +*toxIN* cells. (**D**) Fold coverage from genome sequencing of evolved T4 isolates around *dmd* and *tifA*. (**E**) PCR with divergent primers, indicated in panel (**D**), used to map the size of the repeated unit in the tandem segmental amplifications. (**F**) Schematic of segmental amplification of *dmd-tifA* locus in evo clones highlighting binding of primer pair 2 producing a cross-repeat product that depends on the specific location of the microhomology.

The online version of this article includes the following source data and figure supplement(s) for figure 4:

**Source data 1.** Uncropped and labeled gel for *Figure 4E*.

**Figure supplement 1.** Titers of evolving T4 populations remain roughly constant over 25 rounds of evolution.

**Figure supplement 2.** T4 replicate evolutions overcome *toxIN* using similar segmental amplifications of *tifA*.

**Figure supplement 2—source data 1.** Uncropped and labeled gel for *Figure 4—figure supplement 2A*.

continued the evolution for this and the other four replicate populations for 20 additional rounds, maintaining selection for infection of +*toxIN* cells. By plating serial dilutions of the evolving populations on +*toxIN* lawns, we could track, as they arose, T4 clones that could infect +*toxIN* cells in each population (*Figure 4A–C*, *Figure 4—figure supplement 1*). ToxIN-resistant phages emerged at a different round in each population, but with comparable levels of resistance stabilizing after 25 rounds (*Figure 4A ,B*, *Figure 4—figure supplement 1*). We then sequenced the genome of one phage clone isolated from each population (*Figure 4C*). There were no point mutations in common across all five evolved phage clones. However, as with the first clones sequenced (*Figure 1D*), each of these sequenced genomes had an amplification of the *dmd-tifA* locus, as manifested by an approximately tenfold increase in read counts in this region (*Figure 4D*, *Figure 4—figure supplement 2A*). For the clone from population 5, *dmd* is included in the amplification but not the promoter upstream. However, in this case there is a promoter downstream of *tifA* that ends up oriented in a manner that would allow it to drive expression of the *dmd-tifA* cistron following amplification (*Figure 4—figure supplement 2B*).

The precise boundaries of the segmental amplification differed in each evolved population. This finding indicates that the same adaptive solution, the amplification of the *dmd-tifA* locus, occurred independently in each evolved population. We performed a PCR with divergent primers that would only amplify a product following a segmental amplification to confirm the size of the amplified region in each population (*Figure 4D–F*). The different sizes of the band in clones from different populations confirmed that the repeat region was unique in clones of each population (*Figure 4—figure supplement 2A*). Sanger sequencing of the PCR products revealed that the boundaries of each segmental amplification coincided with unique, short (typically 6–10 bp) regions of microhomology that normally flank the *dmd-tifA* locus (*Figure 4D*, *Figure 4—figure supplement 2C*). In the evolved clones, only one of the two regions of microhomology remain between neighboring repeats (*Figure 4—figure supplement 2C*). Thus, the segmental amplifications observed likely arose through homologous recombination between these regions of microhomology, as also seen previously in T4 (*Kumagai et al., 1993*; *Mosig, 1987*; *Wu et al., 1991*; *Wu et al., 2021*).

Strikingly, in addition to an amplification of the *dmd-tifA* locus, each evolved clone of T4 also had large deletions elsewhere in its genome (*Figure 5A and B*). Individual deletions were different in each clone and ranged from 0.7 to 6 kb in length, for a total of 5–11 kb lost in each genome (*Figure 5C*). We confirmed the absence of each deletion by PCR using primers that flank the deleted region inferred by genome sequencing (*Figure 5—figure supplement 1A*). These deletions likely enable T4 to maintain a constant genome size by acquiring deletions that compensate for the amplifications selected during evolution. The T4 genome is packaged by a headful mechanism in which 172 kb from a genomic concatemer is inserted into a pro-head. The inserted DNA represents 102% of the T4 genome (168 kb), with ~3.3 kb of terminal redundancy (*Grossi et al., 1983*; *Kim and Davidson, 1974*; *Rao and Black, 2005*). To ensure stable packaging and inheritance of the genome, the segmental amplifications that arose following selection on +*toxIN* cells likely required the genome to contract elsewhere. Consistent with this interpretation, the size of the deleted region in each evolved clone was roughly proportional to the size of the segmental amplification, as estimated from the increased read coverage across the *dmd-tifA* region (*Figure 5C*).

Our results indicate that T4 rapidly evolves through recombination between short regions of microhomology within its genome (*Kumagai et al., 1993*; *Mosig, 1987*; *Wu et al., 2021*). To overcome the *toxIN* system, such recombination events cause the segmental amplification of *dmd* and *tifA*. The resulting genome is then likely too big to be stably maintained, giving rise to the variability in genome coverage seen during intermediate rounds of our evolution, with different individual plaques containing different deletions that restore genome size (*Figure 1—figure supplement 2A*, *Figure 5—figure supplement 1B*). At later rounds, one of these genome configurations rises to a high frequency in the population (*Figure 5*, *Figure 5—figure supplement 1B, C*).

## Genome deletions prevent T4 from infecting other strains of *E. coli*

None of the 62 genes that are essential for producing new virions during infection of *E. coli* in rich media at 37°C (*Figure 6—figure supplement 1A*) were included in the deletions identified in our evolved clones (*Miller et al., 2003*). However, we hypothesized that the genes lost in our evolved clones may compromise infection of other strains of *E. coli*.

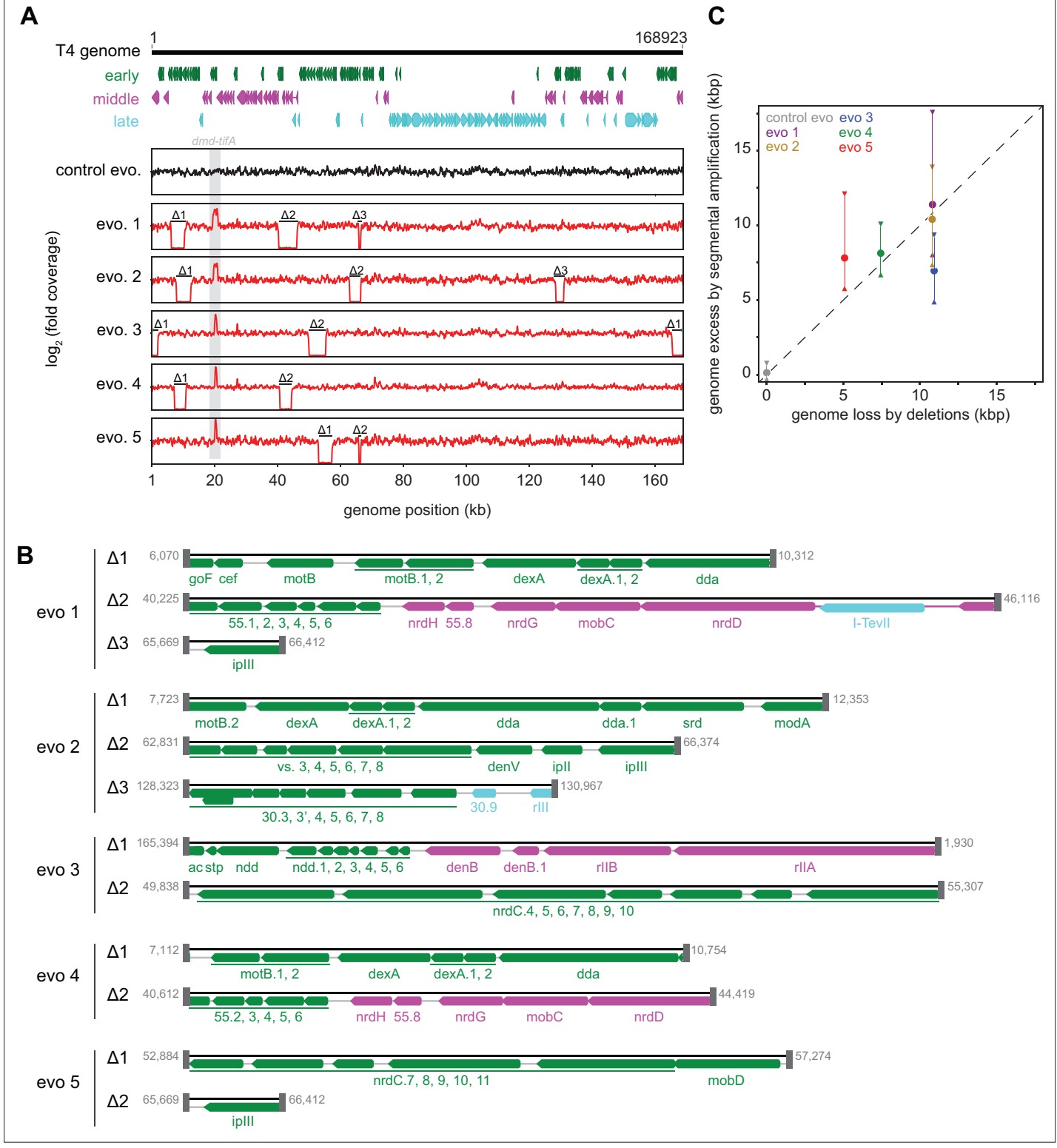

**Figure 5.** Segmental amplifications in the T4 genome are compensated by large genomic deletions. (**A**) Fold coverage calculated from genome sequencing of clones from replicate evolved populations plotted across the entire genome, showing regions of increased coverage (*dmd-tifA* locus) and large genomic deletions manifesting as a loss of coverage. Genes are colored based on their regulation during the phage life cycle; early (green), middle (magenta), and late (cyan). (**B**) Summary of deleted regions in each evolved clone showing the flanking regions of microhomology (gray rectangles). (**C**) Scatter plot of increase in genome size estimated from fold coverage across *dmd-tifA* repeat region (circle represents median and triangles represent interquartile range of coverage) against the size of genome deletion in clones from each population.

*Figure 5 continued on next page*

*Figure 5 continued*

The online version of this article includes the following source data and figure supplement(s) for figure 5:

**Figure supplement 1.** Genome deletions fix over the course of the evolution to compensate for amplifications at the *dmd-tifA* locus.

**Figure supplement 1—source data 1.** Uncropped and labeled gel for *Figure 5—figure supplement 1A*, genome deletions.

**Figure supplement 1—source data 2.** Uncropped and labeled gel for *Figure 5—figure supplement 1C*, left (evo 3).

**Figure supplement 1—source data 3.** Uncropped and labeled gel for *Figure 5—figure supplement 1C*, right (evo 5).

One of the deletions that arose in evolved phage population 3 (evo 3) contained the *rII* locus (*Figure 5B*). The *rIIA* and *rIIB* genes allow T4 to replicate in *E. coli* strains harboring a $\lambda$-lysogen (*Benzer, 1955*). The RexAB system of $\lambda$ allows it to exclude T4 *rII* mutants by a mechanism that remains poorly characterized (*Wong et al., 2021*). As predicted, the T4 clones that have deletions spanning *rII* lost the ability to replicate on a $\lambda$-lysogen of MG1655 (*Figure 6A*, *Figure 6—figure supplement 1B*). Spot assays and whole-genome sequencing of clones isolated from different time points during the evolution confirmed that this population of phage first acquired the ability to infect *+toxIN* cells and then subsequently lost the ability to replicate in a $\lambda$-lysogen once the genomic deletions including *rII* had fixed in the population (*Figure 5—figure supplement 1A–C*, *Figure 6—figure supplement 1B*). Expressing *rIIA* and *rIIB* from plasmids in a MG1655 $\lambda$-lysogen partially restored the infectivity of an evolved T4 clone, confirming the causal nature of the *rII* deletion (*Figure 6A*). These results suggest that the genes lost to compensate for *dmd-tifA* amplification may include genes that help T4 overcome other defense and exclusion mechanisms, like RexAB.

The *rII* deletions were only found in clones from one of the five evolved populations and specifically affect infection of $\lambda$-lysogens. We therefore tested the evolved T4 clones for their ability to infect strains from the ECOR collection, a set of *E. coli* strains isolated from diverse environments that have similar core genomes but widely varying accessory genomes (*Ochman and Selander, 1984*). We focused on strains that can be infected by our ancestral T4 phage, which included four strains from this collection, ECOR13, ECOR16, ECOR17, and ECOR71, as well as *E. coli* str. C (*Figure 6B*, *Figure 6—figure supplement 1C*). Clones from populations 1, 2, 3, and 5 (evo 1, 2, 3, and 5) failed to infect ECOR17 and clones from populations 1 and 3 (evo 1 and 3) failed to infect ECOR71 (*Figure 6B*). The loss-of-replication phenotype arose in each population after they had acquired the ability to overcome *toxIN*, indicating that deletions that compensate for the segmental amplifications led to the loss of genes required for infecting these hosts (*Figure 6—figure supplement 1B and D*).

To identify the gene(s) responsible for loss-of-replication, we asked which phage gene deletions correlated with the inability to infect ECOR17. The only gene lost in common across clones in evo 1, 2, and 5 was *ipIII*, implying that it enables T4 to infect ECOR17 (*Figure 5B* and *Figure 6C*). T-even phages encode a set of proteins – including IPIII – called internal proteins: they contain a sequence motif that drives packaging into mature virions such that the proteins are injected into a host with the phage genome (*Kutter et al., 1995*; *Leiman et al., 2003*; *Mullaney and Black, 1996*). IPI inhibits the restriction-modification system *gmrSD* (*Bair and Black, 2007*), but the function of the other internal proteins is not known. We hypothesized that IPIII may also be an anti-defense protein that targets a host defense system. Indeed, providing *ipIII*$_{T4}$ on a plasmid in ECOR17 restored the ability of clones from evo 2 and 5 to infect this host (*Figure 6D*). We also found that having the host cells express a variant of IPIII lacking the capsid-targeting sequence (*ipIII*$_{T4}$$\Delta$*CTS*) restored infectivity of clones from evo 2 and 5, indicating that IPIII functions in the cytoplasm of ECOR17 to enable T4 replication (*Figure 6—figure supplement 1E*). For the clone from evo 1, *ipIII* on a plasmid did not restore infectivity, suggesting that phages in this population lack an additional gene required to infect ECOR17.

To identify the relevant host defense systems in ECOR17, we used available annotation pipelines (*Payne et al., 2021*; *Tesson et al., 2022*) to predict phage-defensive operons in the ECOR17 genome, finding six potential systems, which we deleted, individually, from ECOR17 (*Figure 6—figure supplement 1F*). None of these deletions restored the infectivity of phage that lack IPIII, suggesting that an unknown defense system exists in ECOR17 and that IPIII enables T4 to overcome this system. Though the evo 3 clone also lost the ability to infect ECOR17, its deletions do not include IPIII. For this clone, a disruption of *dsr1* in ECOR17 restored infection (*Figure 6E*, *Figure 6—figure supplement 1F*). Dsr1 contains a Sirtuin domain and likely blocks phage replication by disrupting cytoplasmic NAD+ levels (*Garb et al., 2021*). Expression of the ECOR17 *dsr1* gene from its native promoter on a medium-copy

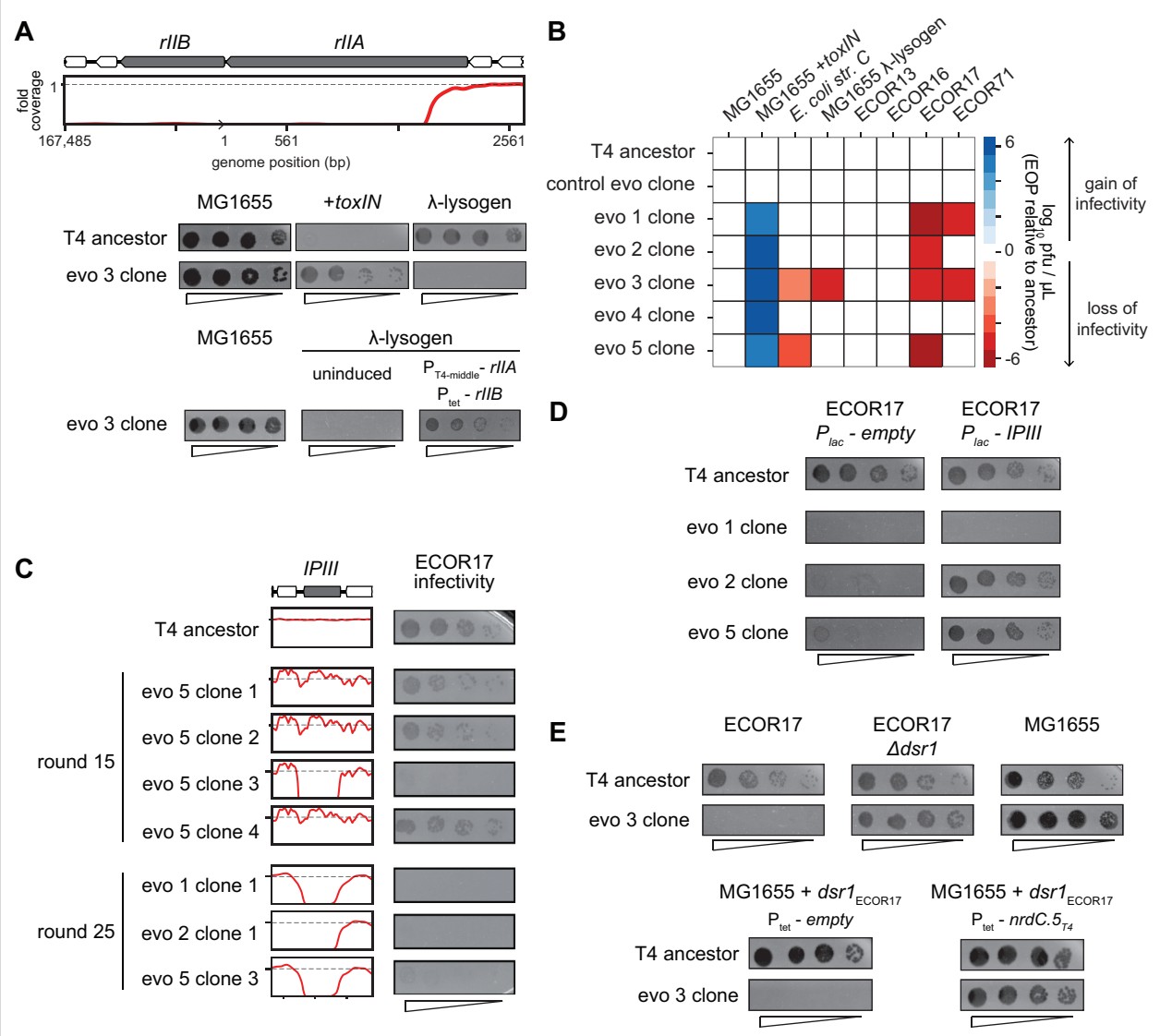

**Figure 6.** T4 genes lost during evolution are essential for replication in alternate hosts. (**A**) (Top) Fold coverage map for *rIIA* and *rIIB* locus in a clone from population 3 (evo 3). (Bottom) Serial dilutions of the T4 ancestor and evo 3 clone 1 on the strains indicated. (**B**) Heat map quantifying the relative efficiency of plaquing (EOP) of each evolved clone compared to the T4 ancestor for the indicated *E. coli* strains evaluated as the average of three biological replicates. (**C**) Fold coverage maps for the *IPIII* gene in the indicated clones of T4 from evolved populations evo 1, 2, and 5, with the corresponding serial dilutions of each clone on ECOR17. (**D**) Serial dilutions of evolved isolates with *IPIII* deletions on ECOR17 and ECOR17 +*IPIII* cells. (**E**) (Top) Serial dilutions of T4 ancestor and evo 3 clone 1 on ECOR17, ECOR17 Δ*dsr1*, and MG1655 lawns. (Bottom) Serial dilutions of T4 ancestor and evo 3 clone 1 on MG1655 with plasmid-borne *dsr1*$_{ECOR17}$ also harboring a plasmid expressing *nrdC.5*$_{T4}$ or empty vector.

The online version of this article includes the following figure supplement(s) for figure 6:

**Figure supplement 1.** Evolved clones of T4 fix genomic deletions that contain genes essential for infecting alternate hosts.

plasmid conferred protection to MG1655 against the evo 3 clone (***Figure 6E***). Thus, a T4 gene deleted in the evo 3 clone must overcome Dsr1 in the context of T4 ancestor infection. Expressing the uncharacterized T4 early gene *nrdC.5* under an inducible promoter restored infectivity of the evo 3 clone on MG1655 carrying *dsr1* (***Figure 6E***). Thus, the T4 genome contains multiple genes (including *ipIII* and *nrdC.5*) that enable the infection of multiple hosts by overcoming the suite of defense systems in these hosts. Our results support the notion that phage genomes are flexible and can rapidly amplify genetic material allowing them to overcome a specific defense system, but at the cost of losing genes that enable infection of other hosts (***Figure 6B***).

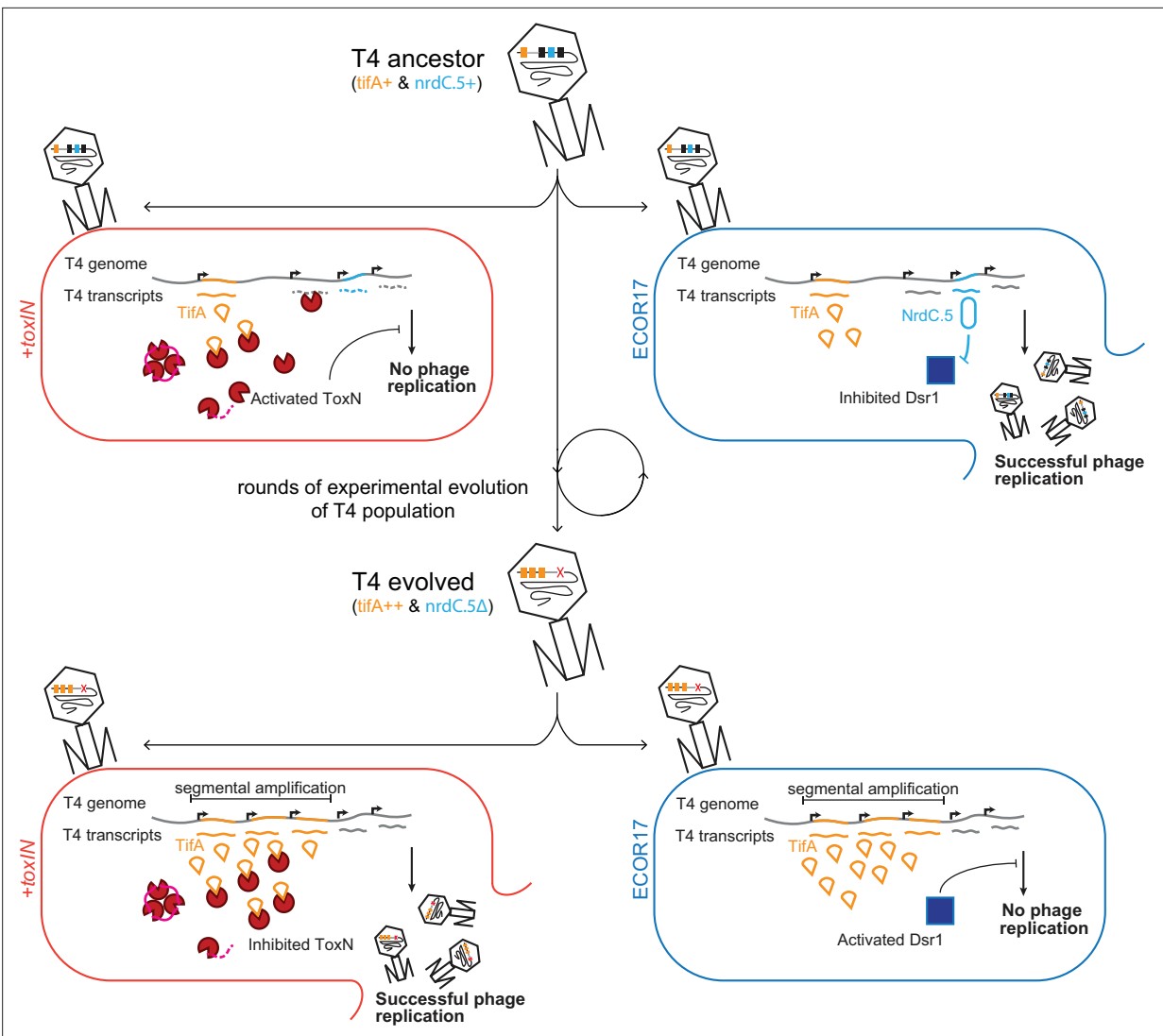

**Figure 7.** Model of T4 evolving to overcome *toxIN* by genomic variation affecting host range. The genome of T4 contains a number of accessory genes (*tifA*, *nrdC.5*, etc.) that enable infection of hosts by overcoming hosts defenses (*toxIN*, *dsr1*, etc.). T4 ancestor is unable to infect +*toxIN* cells due to inability to inhibit the action of ToxN but can infect an alternate host (ECOR17) using *nrdC.5* to overcome the bacterial defense gene *dsr1*. Experimental evolution to infect +*toxIN* cells produce T4 evolved clones that have segmental amplifications of the tifA locus and compensatory deletions of nonessential genes like accessory genes needed to infect alternate hosts. A T4 evolved clone can thus infect +*toxIN* cells by having enough TifA to inhibit ToxN, but has lost *nrdC.5* in a deletion, and thus lost the ability to infect ECOR17.

## Discussion
### TifA and an anti-TA island in T4-like phage

Bacteria and phages are locked in an ongoing coevolutionary battle, with extant genomes of each reflecting the outcomes of the underlying molecular arms race (*Hampton et al., 2020*; *Koskella and Brockhurst, 2014*). Given the fast generation times of bacteria and phages, experimental evolution offers an opportunity to study this arms race as it unfolds. Here, using an adapted version of Appelmans protocol (*Appelmans, 1921*; *Burrowes et al., 2019*; *Mapes et al., 2016*), we evolved T4 to overcome the type III TA system *toxIN*. Genome sequencing of evolved clones identified a two-gene operon *dmd-tifA* that was independently amplified in five replicate populations. We demonstrated that *tifA*, which encodes an 85 amino acid protein, inhibits ToxN, and that the amplification of *tifA* allows T4 to overcome plasmid-borne *toxIN* (*Figure 7*). Homologs of TifA are found in many T4-like phage, suggesting that these phages may frequently encounter hosts harboring *toxIN*-like systems.

However, these homologs vary in both their sequence and their ability to inhibit ToxN, suggesting that each TifA may be 'tuned' to inhibit a particular, cognate ToxN of an ecologically relevant host.

TifA is analogous to the co-operonic gene *dmd*, which encodes a direct inhibitor of RnlA, the toxin of the *rnlAB* system in *E. coli*. Additionally, the nearby and conserved gene *61.2* (now *adfA*) was recently found to overcome the type IV TA system *darTG* (*LeRoux et al., 2022*). Thus, this region of the T4 genome may be an anti-TA island, similar to the anti-RM and anti-CRISPR islands found in other phages (*Pinilla-Redondo et al., 2020*). Other genes in this region of the T4 genome are attractive anti-toxin candidates.

## Recombination-mediated amplifications and deletions drive T4 genome evolution

T4 evolved to overcome *toxIN* by amplifying *dmd-tifA*, likely via recombination between short homologous sequences that flank this locus. These recombination events may arise during replication of the T4 genome or during the repair of DNA breaks. The unique amplifications that arose independently in our five replicates highlight the high rates of recombination-based mutagenesis in phage evolution. Similar recombination events between short (2–10 bp) regions of homology in T4 have been seen previously in studies of the *rII* locus (*Kumagai et al., 1993*), genes *16–18* (*Wu et al., 1991*), and escape from CRISPR-Cas defense (*Wu et al., 2021*).

Our work also indicates that T4 can undergo large genomic changes through recombination between short regions of homology, leading to loss of the intervening region (*Figure 5A*, *Figure 5—figure supplement 1A*). Genome packaging in T4 proceeds via a headful mechanism (*Rao and Black, 2005*), so any substantial gain in DNA must be offset by comparably sized losses. Thus, the gains in genetic material resulting from amplification of *dmd-tifA* likely necessitated the loss of other, nonessential DNA to maintain a genome that could be fully packaged into the head. A similar pattern of gene amplification in the *16–18* region of T4 followed by deletion of the gene *alt* was previously reported (*Wu et al., 1991*).

Gene amplification is a major source of evolutionary innovation. For instance, antibiotic resistance in bacteria frequently arises through gene amplification. Seminal studies of *Salmonella* demonstrated that spontaneous tandem duplications arise frequently in bacterial genomes and can be present in as much as 3% of a population (*Anderson and Roth, 1981*; *Anderson and Roth, 1977*). Gene duplications can provide intrinsic advantages, as with antibiotic resistance genes or in our case of *tifA*. Additionally, duplications create additional copies of a gene and hence provide a larger target for point mutations; if a beneficial mutation arises in any copy of the gene, it can provide a selective advantage and be maintained while other copies are lost by subsequent recombination. Such a scenario explains the apparently high rates of *lac-* reversions measured by Cairns and Foster when selecting for restoration of LacZ activity in strains harboring a point mutation in *lacZ* (*Andersson et al., 1998*; *Cairns and Foster, 1991*; *Hendrickson et al., 2002*). Similarly, a selection for poxviruses that can overcome the host restriction factor protein kinase R revealed an initial segmental amplification of the anti-host factor K3L, followed by subsequent acquisition of a beneficial point mutation in one copy K3L and subsequent loss of additional copies of the gene (*Elde et al., 2012*).

In the case of *tifA*, the entire T4 ancestor population likely contains a single copy of the *dmd-tifA* locus, which is insufficient to overcome plasmid-borne *toxIN*. Thus, the segmental amplifications that increase *tifA* copy number could not have been identified through a direct, one-step selection for escape mutants. However, the recombination events leading to these amplifications arose within several rounds in each of our replicate populations, indicating that this is a readily available pathway to overcoming *toxIN*. We did not observe point mutations in *tifA* early on that improve its inhibition of ToxN, likely because segmental amplifications via recombination arise more frequently. We also did not observe beneficial point mutations arising after the segmental amplifications, leading to subsequent collapse of the amplification, as with protein kinase R. We speculate that the fitness gains and number of beneficial point mutations in *tifA* are likely too small to arise and fix.

## The evolution of phage host range

The genes deleted in our evolved T4 phage are all nonessential for infection of *E. coli* MG1655 grown in laboratory conditions but may be crucial for T4 to infect other hosts. Indeed, we found that the deletions in four of our five replicate populations compromised infection of other strains of *E. coli*.

The loss of *rII* led to an inability to infect $\lambda$ lysogens, consistent with prior work showing that *rII* is required to neutralize the $\lambda$-encoded anti-T4 system RexAB. Similarly, we found that IPIII, which was lost in three populations, is required for T4 to infect *E. coli* strain ECOR17. Internal proteins, including IPIII, are packaged with and then subsequently injected with the T4 genome. These proteins likely act against 'first-response' defense systems like RM systems in each host. For instance, IPI is known to inhibit the *E. coli* type IV RM system *gmrSD*. Which defense system IPIII inhibits remains to be determined.

The accessory genomes of phages likely include a broad range of anti-defense genes that profoundly influence their host ranges. The set of anti-defense genes in a phage genome at a given time will reflect, in part, those that have recently been selected for based on the hosts that phage has infected. As our experimental evolution demonstrates, if selection to infect a host and overcome its defense systems is not maintained, the corresponding anti-defense genes, which are not essential for phage replication, can be lost following pressure to infect new and different hosts (*Figure 7*). Such genome dynamics, which stem from the relentless coevolution of phage with their hosts, may help to explain the staggering diversity of accessory genes within phage genomes (*Pope et al., 2015*) and their sometimes narrow host ranges across hosts with varied and dynamic immune profiles (*Hussain et al., 2021*; *Kauffman et al., 2022*).

## Concluding remarks

Our experimental evolution approach enabled the identification of new anti-phage defense and phage counter-defense mechanisms. In addition to finding TifA as a novel inhibitor of ToxN, the characterization of our evolved T4 isolates indicated that IPIII likely antagonizes a novel, as-yet unknown defense system in ECOR17. Additionally, we discovered a homolog of the recently identified Dsr1 system in ECOR17 that can be counteracted by the nonessential gene *nrdC.5* in wild-type T4. Collectively, these findings underscore how much remains to be discovered about the molecular arms race between bacteria and phages, which has profoundly shaped the genomes of both.

Phages evolve rapidly, but the forces and mechanisms that drive this evolution are poorly understood. Experimental evolution helps to reveal the tempo and mutations critical to phage genome evolution. Efforts to dissect how phages evolve to overcome a range of host barriers, including different defense systems, promise to provide further insights into how phage evolution occurs and the mechanisms responsible. In addition, such studies may also inform practical efforts to engineer phages and develop phage therapeutics.

# Materials and methods

## Strains and growth conditions

All strains, plasmids, and primers used in this study are listed in *Supplementary file 1*. For all phage experiments in liquid media and all phage spotting experiments, *E. coli* MG1655 strains were grown in Luria broth (LB) medium. For pBAD33-ToxN induction experiments, cells were grown in M9 (10× stock made with 64 g/L $Na_2HPO_4$-$7H_2O$, 15 g/L $KH_2PO_4$, 2.5 g/L NaCl, 5.0 g/L $NH_4Cl$) medium supplemented with 0.1% casamino acids, 0.4% glycerol, 0.4% glucose, 2 mM $MgSO_4$, and 0.1 mM $CaCl_2$ (M9-glucose). For plasmid construction, *E. coli* DH5α and TOP10 cells were grown in LB medium. Antibiotics were used at the following concentrations (liquid; plates): carbenicillin (50 µg/mL; 100 µg/mL) and chloramphenicol (20 µg/mL; 30 µg/mL).

## Plasmid construction

### pKVS45-*tifA* and pKVS45-*dmd*

T4 *tifA* and *dmd* were individually PCR-amplified from purified T4 genomic DNA with primer pairs SS-5/SS-6 and SS-7/SS-8, respectively, which also add flanking BamHI and KpnI sites. These PCR products were then cloned into the BamHI and KpnI sites of the anhydrotetracycline (aTc)-inducible pKVS45 vector (pKVS45-*tifA* and pKVS45-*dmd*). pKVS45-*tifA*-FLAG was constructed via round-the-horn PCR on pKVS45-*tifA* using the primer pair SS-9/10. pKVS45-*tifA* ΔATG was constructed via round-the-horn PCR on pKVS45-*tifA* using the primer pairs SS-31/32 followed by blunt-end ligation. pKVS45-*tifA* ablated ToxN-site was constructed via round-the-horn PCR on pKVS45-*tifA* using the primer pairs SS-33/34 followed by Gibson Assembly using the 2× HiFi DNA Assembly Master Mix (NEB E2621).

TifA codons were recoded using Geneious and ordered as a gene-block fragment from IDT (with 46 of 85 codons recoded with 55 nucleotide changes) and cloned into pKVS45 by Gibson Assembly at the same site as pKVS45-*tifA*. For pKVS45-*tifA*-homologs, *tifA* homologs from T2, T6, and RB69 were each PCR-amplified from purified phage genomic DNA with primer pairs SS-11/12, SS-13/14, and SS-15/16, respectively, which also add sequences that overlap the aTc-inducible pKVS45 vector. pKVS45 was digested with BamHI and KpnI, and then the PCR-amplified *tifA* variants were inserted into the digested plasmid using Gibson assembly.

### pEXT20-*tifA*

The *tifA* homologs from T2, T4, T6, and RB69 were each PCR-amplified from purified phage genomic DNA with primer pairs SS-17/18, SS-19/20, SS-21/22, and SS-23/24, respectively, which also add sequences that overlap the isopropylthio-β-galactoside (IPTG)-inducible pEXT20 vector. pEXT20 was digested with SacI and SalI, and then the PCR-amplified *tifA* variants were inserted into the digested plasmid using Gibson Assembly.

### pEXT20-*ipIII*$_{T4}$

The *ipIII*$_{T4}$ gene was PCR-amplified from T4 genomic DNA with primer pair SS-35/36, which add sequences that overlap the IPTG-inducible pEXT20 vector. pEXT20 was digested with SacI and SalI, and *ipIII*$_{T4}$ was inserted into the digested plasmid using Gibson Assembly.

### pBR322-*rIIA*$_{T4}$

The *rIIA* gene including the upstream native promoter was PCR-amplified from T4 genomic DNA with primer pair SS-37/38, which add sequences that overlap with the pBR322 vector. The pBR322 backbone was amplified with primer pair SS-39/40, and the *rIIA* gene was inserted using Gibson Assembly.

### pKVS45-*rIIB*$_{T4}$

The *rIIB* gene (only the coding sequence) was PCR-amplified from T4 genomic DNA with primer pair SS-41/42, which add flanking BamHI and KpnI sites. This PCR product was then cloned into the BamHI and KpnI sites of the aTc-inducible pKVS45 vector.

### pBR322-*Dsr1*$_{ECOR17}$

The *Dsr1*$_{ECOR17}$ gene including the upstream promoter (300 bp) was PCR-amplified from ECOR17 genomic DNA with primer pair SS-43/44, which add sequences that overlap with the pBR322 vector. The pBR322 backbone was amplified with primer pair SS-39/40, and the Dsr1$_{ECOR17}$ gene was inserted using Gibson Assembly.

### pKVS45-*nrdC.5*$_{T4}$

The *nrdC.5*$_{T4}$ gene (only the coding sequence) was PCR-amplified from T4 genomic DNA with primer pair SS-45/46, which add flanking BamHI and KpnI sites. This PCR product was then cloned into the BamHI and KpnI sites of the aTc-inducible pKVS45 vector.

## Strain construction

### *E. coli* MG1655 *attB*$_λ$::*toxIN* and *E. coli* MG1655 *attB*$_λ$::*toxIN(K55A)*

To construct these strains, *toxIN* or *toxIN(K55A)* was inserted into the *E. coli* MG1655 chromosome using the CRIM system (*Haldimann and Wanner, 2001*). The *toxIN* locus, along with its native promoter and transcription terminators, was PCR-amplified from pBR322-*toxIN* with the primer pair SS-25/26. The CRIM plasmid pAH150 was linearized using PCR with SS-27/28 to remove the *araBAD* promoter and *rrnB* T1 and *rrnB* T2 terminators. *toxIN* was then inserted into linearized pAH150 using Gibson Assembly to yield pAH150-*toxIN*. pAH150-*toxIN(K55A)* was then constructed via round-the-horn PCR of pAH150-*toxIN* with SS-29/30. CRIM insertion of these plasmids into the *attB*$_λ$ locus of *E. coli* MG1655 was then performed as previously described (*Haldimann and Wanner, 2001*) using the pINT-ts helper plasmid. Single insertions were confirmed by PCR and Sanger sequencing.

## ECOR17 gene deletions

To construct these strains, the open-reading frames of the candidate defense systems (*RM-typeI*, *RM-typeIII*, *abi2*, *hhe*, *dsr1*, *cas3*) were designed to be replaced by a kanamycin resistance cassette (kan$^R$) using the $\lambda$-red recombinase system (*Datsenko and Wanner, 2000*). Briefly, >70 bp homologous regions flanking the ORFs were added to the resistance cassette and electroporated into ECOR17 containing pKD46 plasmid that expresses the $\lambda$-red recombinase genes. Deletion of the ORFs was confirmed by PCR across the locus.

## Phage T4 *tifA* gene disruptions

T4 mutants were generated using a CRISPR-Cas system for targeted mutagenesis described previously (*Duong et al., 2020*). Briefly, sequences for RNA guides to target Cas9-mediated cleavage in the *tifA* open-reading frame (but nowhere else in the T4 genome) were designed using the toolbox in Geneious Prime 2021.2.2. Guides were chosen based on a high activity score that may have a loose correlation with their efficiency of cutting of the T4 genome (*Duong et al., 2020*). The guides were inserted into the pCas9 plasmid (Addgene #42876) and tested for their ability to target the T4 genome by measuring the restriction of T4 spotting (*Figure 2—figure supplement 2*). Specifically, though wild-type T4 is not strongly restricted by any of the guides, a clone of T4 with a mutation in the β-glucosyltransferase gene, *bgt(II/5-6Δ)*, showed strong sensitivity to one of the guides. We expect the deletion to cause a deficiency in β-glucosyltransferase activity, leading to the loss of β-glucose modification of about 30% of the hydroxymethylcytosines (*Lehman and Pratt, 1960*). pCas9-containing guide *tifA*$_{T4}$-cr4 (AATTCCACTCGACCAAATGGAGG, constructed with primers SS-47/48) was selected because it had the strongest effect on plaquing efficiency of T4(*bgt(II/5-6Δ)*) (*Figure 2—figure supplement 2*). Escape plaques were isolated by plating T4(*bgt(II/5-6Δ)*) at high titer onto a lawn of MG1655 harboring the plasmid pCas9-*tifA*$_{T4}$-cr4. The *tifA* locus from escape plaques was then PCR-amplified and Sanger-sequenced to identify two unique clones that had a 98 bp deletion and a 5 bp insertion in *tifA* that disrupt its coding sequence (*Figure 2—figure supplement 2*). Finally, the clones were isolated to purity and submitted for whole-genome sequencing to confirm that no other mutations were present.

## Phage experiments in solid and liquid media

Phage spotting assays were conducted as described in *Guegler and Laub, 2021* and were conducted at least three times independently, with representative experiments shown in the figures. Briefly, phage stocks isolated from single plaques were propagated in *E. coli* MG1655 at 37°C in LB. To titer phage, stocks were mixed with *E. coli* MG1655 and melted LB + 0.5% agar and spread on LB + 1.2% agar plates at 37°C. For phage spotting assays, the bacterial strain of interest was mixed with LB + 0.5% agar and spread on an LB + 1.2% agar + antibiotic plate. For experiments in which expression of *tifA* was induced with aTc (P$_{tet}$-*tifA*), 100 ng/μL aTc was added to LB + 0.5% agar at a final concentration of 100 ng/mL prior to mixing with bacterial cells. For the experiment inducing *rIIA* and *rIIB*, aTc was added to 10 ng/mL to minimize the toxicity of the *rII* system. For experiments in which expression of genes was induced with IPTG (P$_{lac}$-*genes*), 100 mM IPTG was added to LB + 0.5% agar to a final concentration of 100 μM prior to mixing with bacterial cells. Phage stocks were then serially diluted in 1× FM buffer (20 mM Tris–HCl pH 7.4, 100 mM NaCl, 10 mM MgSO$_4$), and 2–3 μL of each dilution was spotted on the bacterial lawn. Plates were then grown at room temperature or 37°C overnight and plaques quantified the following day. EOP was calculated by comparing the ability of the phage to form plaques on an experimental strain relative to the control strain.

Phage experiments in liquid culture were conducted at 37°C. Single colonies were grown overnight in LB. Overnight cultures were back-diluted to OD$_{600}$ = 0.01 and grown to OD$_{600}$ = 0.3 in fresh LB such that cells re-entered exponential growth. Cultures were then back-diluted to OD$_{600}$ = 0.01 in fresh LB, and phage was added to each culture. Growth following phage infection was measured in 24-well plates at 15 min intervals with orbital shaking at 37°C on a plate reader (BioTek). Data points reported are the mean of three independent biological replicate growth curve experiments. Raw data are provided on GitHub.

## High-throughput evolution of T4 to infect resistant *E. coli*

Phage populations are evolved to infect resistant hosts using a modification of the Appelmans protocol originally developed to generate phage therapy cocktails (*Appelmans, 1921*). This protocol was more recently used to modify the host range of phage (*Burrowes et al., 2019*; *Mapes et al., 2016*). Briefly, a population of phage is propagated on a sensitive host to maintain phage population size and generate genetic diversity, and also a resistant host to select for phage that can infect the novel host. In our experiment, the resistant host was MG1655 with a *toxIN*-containing plasmid (ML3328), and the sensitive host was MG1655 with an empty plasmid (ML3330). Thus, T4 needs to overcome the defense conferred by *toxIN* to successfully invade the resistant host. To visually track the progress of evolution, the evolution experiment was setup in 96-well plates with phage populations inoculated with host cells at a range of MOI (*Figure 1—figure supplement 1*). A phage blank well was also maintained to confirm there is no cross-contamination during the setup of each round of evolution.

T4 ($10^6$ pfu/μL) lysate prepared in MG1655 was then serial-diluted tenfold in FM buffer to generate inoculum samples from $10^6$ to 1 pfu/μL. 10 μL of each the phage inoculums were added to the corresponding wells in a row. Each well contains 100 μL of supplemented LB (LB + 15 μg/mL L-tryptophan + 5 μg/mL thiamine) with $10^5$ bacterial cells. Bacterial culture was back-diluted from overnight growth of single colonies in LB (or M9) at 37°C using $OD_{600}$ of 1 as an estimated cell density of $7 \times 10^8$ cells/mL. The bacterial cells are ML3328 (*+toxIN*) and ML3330 (*-toxIN*) in two separate rows. The first wells of each row are inoculated with only 10 μL of FM buffer as a no-phage control check against phage cross-contamination during the setup of evolution plate. The second to eighth wells of each row receive inoculum from the phage serial dilutions to setup wells with phage-to-bacteria rations (MOI) of $10^2$ to $10^{-4}$. Each pair of rows correspond to an independent evolving population, with the top two rows in the plate being a control evolution with T4 infecting two rows of ML3330 to identify mutations that arise when T4 is passaged on just MG1655 with no *toxIN*. The remaining rows on the plate accommodate five independent evolving populations of T4. For the first round, the same clonal population of T4 is used to start all the evolving populations. After inoculating the cell culture with phage, the evolution plate is sealed with breathable film and incubated at 37°C on a plate shaker at 1000 RPM overnight (16– 20 hr). The plate is harvested by pooling all cleared wells and the first uncleared well (up to a total of five wells per row from the highest MOI) in every pair of rows to generate the population of phage at the end of the round. In a round of evolution, phage in the highest MOI well lyse all cells in a single infection generation, while at the lowest MOI (one phage particle inoculated into $10^5$ bacteria) about three rounds of phage infection (with an estimated T4 burst of 100) are needed to lyse all the bacteria. Thus, a phage population after a round of evolution contains particles that have gone through 1–3 infection generations from a phage at the start of the round. The cell debris and unlysed cells in pooled samples were pelleted (4000 RPM at 4°C for 25 min) and the supernatant phage lysate was transferred to 96-deep-well blocks for long-term storage (with 40 μL chloroform to prevent bacterial growth). Pooled samples generated six populations – five replicate populations evolving on ML3328 and ML3330, and one control population evolving on ML3330 alone. Twenty-five rounds of evolution were performed saving a total of 150 populations (six populations at each of 25 rounds of evolution). The evolution was performed in supplemented LB for the first 18 rounds and was then switched to supplemented M9 for 7 rounds to increase the selective pressure of *toxIN* on T4 replication. Evolving populations were stored as lysates at 4°C, and also as glycerol stocks of infected hosts snap-frozen and left at –80°C.

Evolved clones were isolated by plating to single plaques from the evolved populations on lawns of ML3328 for T4 that can overcome *toxIN*. These evolved clones were saved as phage lysates at 4°C and as glycerol stocks of infected hosts for long-term storage.

## ToxN and TifA overexpression plating assays on solid media

Overexpression plating assays were conducted as described in *Guegler and Laub, 2021*. Briefly, single colonies were grown to saturation overnight in M9-glucose. 1 mL of each overnight culture was then pelleted by centrifugation at $4000 \times g$ for 5 min, washed twice in 1× phosphate-buffered saline (PBS), and resuspended in 500 μL 1× PBS. Cultures were then serially diluted tenfold in 1× PBS and spotted on M9L plates (M9-glycerol supplemented with 5% LB [v/v]) further supplemented with 0.4% glucose (toxin-repressing), 0.2% arabinose (toxin inducing), or 0.2% arabinose and 100 μM IPTG (toxin

and antitoxin inducing). Plates were then incubated at 37°C for 48 hr before imaging. Images shown are representative of at least three independent biological replicates.

## ToxN and TifA overexpression growth curves in liquid culture

Single colonies of *E. coli* MG1655 harboring pBAD33-*toxN* and pEXT20-*tifA_{T4}* were grown to saturation overnight in M9-glucose. These cultures were then back-diluted 100-fold in fresh M9-glycerol supplemented with 0.8% glucose and grown to $OD_{600}$ ~ 0.3 at 37°C in a shaking water bath; additional glucose was added to these cultures to suppress leaky toxicity of ToxN. Cells were pelleted by centrifugation at 4°C and 4000 × *g* for 5 min. Pellets were washed twice in ice-cold M9-glycerol and then resuspended in fresh M9-glycerol. Cells were then back-diluted in fresh M9-glycerol to $OD_{600}$ = 0.2; for cultures in which TifA expression was induced, 100 µM IPTG was also added at this point. Cultures were then grown for an additional 45 min at 37°C in a shaking water bath before being back-diluted to $OD_{600}$ = 0.1 in fresh M9-glycerol. To these back-diluted cultures, 100 µM IPTG and 0.2% arabinose were added to induce TifA and ToxN, respectively, when appropriate. Growth was then measured in 24-well plates at 15 min intervals with orbital shaking at 37°C on a plate reader (Bio-Tek). Data reported are the mean of three technical replicates each of four independent growth curve experiments.

## ToxN–TifA interaction assays with co-immunoprecipitation (co-IP)

50 mL cultures of *E. coli* MG1655 containing pBR322-*toxIN*(+/-His_6) and pKVS45-*tifA*(+/-FLAG) were grown to $OD_{600}$ = 0.4 at 37°C in LB supplemented with 15 µg/mL ʟ-tryptophan and 1 µg/mL thiamine in a water bath shaker and infected with T4 at MOI = 5. To induce expression of TifA, aTc was added to each culture at a final concentration of 100 ng/mL 30 min prior to phage infection. Then, 15 min post-T4 infection, 40 mL of cells from each culture were pelleted by centrifugation at 4000 × *g* for 5 min at 4°C. Pellets were then flash-frozen and stored at –80°C.

For anti-FLAG co-IP experiments, each cell pellet was resuspended in 500 µL of lysis buffer (25 mM Tris–HCl pH 8, 150 mM NaCl, 1 mM EDTA pH 8, 1% Triton X-100, cOmplete Protease Inhibitor Cocktail [Roche], 1 µL/mL ReadyLyse [Novagen], 1 µL/mL DNase I [TakaraBio], 5% glycerol). Cells were lysed by two freeze–thaw cycles, followed by incubation at 30°C for 1 hr. Lysates were clarified by centrifugation at 14,000 × *g* for 10 min at 4°C and then incubated with anti-FLAG-M2 magnetic beads (Sigma) on an end-to-end rotor for 1 hr at 4°C. Beads were then washed three times with 400 µL wash buffer (25 mM Tris–HCl pH 8, 150 mM NaCl, 1 mM EDTA pH 8, cOmplete Protease Inhibitor Cocktail [Roche], 5% glycerol). To elute protein, beads were incubated with 40 µL of 1× Laemmli buffer, vortexed briefly, and boiled at 95°C for 5 min.

Samples from co-IP experiments were analyzed by Western blot as described previously (*Guegler and Laub, 2021*). Briefly, samples were resolved by 4–20% SDS-PAGE and transferred to a PVDF membrane. Membranes were then incubated with rabbit anti-His_6 and anti-FLAG primary antibodies (from Abcam and Cell Signaling Technologies, respectively) at a final concentration of 1:5000 to avoid cross-reactivity with the anti-FLAG-M2 beads. Next, membranes were incubated with horseradish peroxidase (HRP)-conjugated anti-rabbit secondary antibodies (Thermo Fisher) at a final concentration of 1:5000. Blots were developed using the SuperSignal West Femto Maximum Sensitivity Substrate (Thermo Fisher Scientific) and imaged with a FluorChem R Imager (ProteinSimple). Blots shown are representative of three independent replicates.

## Phage genome isolation, Illumina library preparation, and analysis

Phage genomes were prepared by treating high titer lysates (>10^6 pfu/µL) with DNAse I (0.001 U/µL) and RNAse A (0.05 mg/mL) at 37°C for 30 min to hydrolyze any host nucleic acid that remains in the lysate. EDTA is added to a final of 10 mM to inactivate nucleases. The lysate is then incubated with Proteinase K at 50°C for 30 min to disrupt phage capsids and release genomes. The phage genomes are recovered by ethanol precipitation. Briefly, NaOAc pH 5.2 was added to 300 mM followed by 100% ethanol to a final volume fraction of 70%. The samples were left at –80°C for 2 hr to precipitate genomic material. The genomic material was pelleted at 21,000 × *g* for 15 min, and the supernatant is discarded. The pellet was washed in series with 100 µL isopropanol, and 200 µL 70% (v/v) ethanol by pelleting and discarding supernatants for each wash. The pellet was air-dried at room temperature for 30 min and then resuspended in 20–100 µL TE (10 mM Tris–HCl, 0.1 mM EDTA, pH 8) overnight at

room temperature. Genomic material was resuspended by flicking the Eppendorf, and concentration was measured using a NanoDrop spectrophotometer.

Illumina sequencing libraries were prepared starting from 100 to 200 ng of genomic DNA (gDNA). Briefly, the gDNA was sheared by sonication in a Bioruptor. Fragmented gDNA was purified using AmpureXP beads, followed by sequential enzymatic reactions for end repair, 3′ adenylation, and adapter ligation. Barcodes were added to both 5′ and 3′ ends by PCR-amplifying with primers that anneal to the Illumina PE adapters. The libraries were cleaned by Ampure XP clean up using a double cut to elute fragment sizes matching the read lengths of the sequencing run. Libraries were sequenced on either an Illumina MiSeq or NextSeq at the MIT BioMicroCenter after qPCR and fragment analyzer quality control for concentration and size distribution.

Illumina reads were analyzed using custom Python scripts provided on GitHub along with assembled reference genomes. Briefly, the reads were cleaned of adapter sequences using Cutadapt before they are aligned to a reference genome using BWA and SAMtools. Per-position coverage maps were generated using SAMtools depth, and variant single-nucleotide polymorphisms were called using BCFtools. Coverage maps in figures were plotted by using a moving window average of per-nucleotide coverage, with the window size defined as the read length of sequencing run. The ancestral T4 was sequenced to generate a reference genome against which all evolved clones were compared. Illumina reads were also assembled to reference genomes using the assembly pipeline in Geneious Prime 2021.2.2 for cross-checks and visual inspection.

## Homology search and alignment of TifA

Homologs of TifA were first identified by jackhmmer using a Hidden Markov Model built on a small subset of TifA homologs among reference proteomes on UniProt. The overwhelming majority of homologs with full query coverage were in viruses that are members of the phage clade Caudovirales. These 57 sequences were aligned using MUSCLE (with default settings), and a maximum likelihood phylogenetic tree was constructed with FastTree. The Tree separated TifA homologs from different phage genomes into clusters by similarity (color coded, *Figure 3—figure supplement 1*).

## Acknowledgements

We thank A Murray, K Gozzi, I Frumkin, C Vassallo, and T Zhang for comments on the manuscript, S Jones and C Eickmann for assistance in constructing plasmids, and all members of the Laub lab for helpful discussions. We thank the MIT BioMicro Center and its staff for their support in sequencing. This work was supported by an NSF predoctoral graduate fellowship to CKG. MTL is an Investigator of the Howard Hughes Medical Institute.

## Additional information

### Competing interests

Michael T Laub: Reviewing editor, *eLife*. The other authors declare that no competing interests exist.

### Funding

| Funder | Grant reference number | Author |
|---|---|---|
| Howard Hughes Medical Institute | | Michael T Laub |
| National Science Foundation | Graduate Research Fellowship | Chantal K Guegler |

The funders had no role in study design, data collection and interpretation, or the decision to submit the work for publication.

### Author contributions

Sriram Srikant, Chantal K Guegler, Conceptualization, Resources, Data curation, Software, Formal analysis, Validation, Investigation, Visualization, Methodology, Writing - original draft, Writing - review

and editing; Michael T Laub, Conceptualization, Supervision, Funding acquisition, Investigation, Writing - original draft, Project administration, Writing - review and editing

### Author ORCIDs
Sriram Srikant (iD) http://orcid.org/0000-0003-3904-0336
Chantal K Guegler (iD) http://orcid.org/0000-0003-0818-4222
Michael T Laub (iD) http://orcid.org/0000-0002-8288-7607

### Decision letter and Author response
Decision letter https://doi.org/10.7554/eLife.79549.sa1
Author response https://doi.org/10.7554/eLife.79549.sa2

---

## Additional files

### Supplementary files
• Supplementary file 1. List of (a) strains; (b) plasmids; (c) primers; and (d) antibodies.

• MDAR checklist

• Source data 1. Source data folder contains individual raw source images of all gels and western blots presented in the paper.

### Data availability
DNA sequencing data is available at SRA (BioProject ID: PRJNA824875). Scripts used for sequencing data processing and analysis are available at https://github.com/sriramsrikant/2022_T4-toxIN-evo, (copy archived at swh:1:rev:4c0ba4f8ca73988717d1478e3fde7f44378c1b02). Raw data for relevant figures also available at Github.

The following dataset was generated:

| Author(s) | Year | Dataset title | Dataset URL | Database and Identifier |
|---|---|---|---|---|
| Srikant S, Guegler CK, Laub MT | 2022 | Experimental evolution of T4 to overcome *E. coli* toxIN | https://www.ncbi.nlm.nih.gov/bioproject/PRJNA824875/ | NCBI BioProject, PRJNA824875 |

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

# Appendix 1

## Appendix 1—key resources table

| Reagent type (species) or resource | Designation | Source or reference | Identifiers | Additional information |
|---|---|---|---|---|
| Strain, strain background (*Escherichia coli*) | DH5a | Invitrogen | N/A | Cloning strain |
| Strain, strain background (*E. coli*) | MG1655 | Maintained by Laub lab | ML-6 | N/A |
| Strain, strain background (*E. coli*) | C (13706) | Félix d'Hérelle Reference Center for Bacterial Viruses, Université Laval | HER #1036; ML-3793 | Original source: ATCC |
| Strain, strain background (*E. coli*) | ECOR17 | Thomas S. Whittam STEC Center at Michigan State University | ML-3794 | N/A |
| Strain, strain background (*E. coli*) | ECOR17 Δ*RM-typeI* | This study | ML-3797 | Derivative of ML-3794 |
| Strain, strain background (*E. coli*) | ECOR17 Δ*RM-typeIII* | This study | ML-3798 | Derivative of ML-3794 |
| Strain, strain background (*E. coli*) | ECOR17 Δ*abi2* | This study | ML-3799 | Derivative of ML-3794 |
| Strain, strain background (*E. coli*) | ECOR17 Δ*dsr1* | This study | ML-3800 | Derivative of ML-3794 |
| Strain, strain background (*E. coli*) | ECOR17 Δ*hhe* | This study | ML-3801 | Derivative of ML-3794 |
| Strain, strain background (*E. coli*) | ECOR17 Δ*cas3* | This study | ML-3802 | Derivative of ML-3794 |
| Strain, strain background (*E. coli*) | ECOR71 | Thomas S. Whittam STEC Center at Michigan State University | ML-3803 | N/A |
| Strain, strain background (*E. coli*) | ECOR13 | Thomas S. Whittam STEC Center at Michigan State University | ML-3804 | N/A |
| Strain, strain background (*E. coli*) | ECOR16 | Thomas S. Whittam STEC Center at Michigan State University | ML-3805 | N/A |
| Strain, strain background (*E. coli*) | MG1655 $\lambda$-lysogen | This study | ML-3971 | Derivative of ML-6 |
| Strain, strain background (*E. coli*) | MG1655 *attB*::*toxIN* | This study | ML-3789 | Derivative of ML-6 |
| Strain, strain background (*E. coli*) | MG1655 *attB*::*toxI-toxN(K55A)* | This study | ML-3790 | Derivative of ML-6 |
| Strain, strain background (T4) | T4 ancestor | Guegler and Laub, 2021 | phML-31 | Maintained by Laub lab |
| Strain, strain background (T4) | T4 control evo round 25 clone 1 | This study | phML-32 | Derivative of phML-31 |
| Strain, strain background (T4) | T4 evo 1 round 25 clone 1 | This study | phML-33 | Derivative of phML-31 |
| Strain, strain background (T4) | T4 evo 2 round 25 clone 1 | This study | phML-34 | Derivative of phML-31 |
| Strain, strain background (T4) | T4 evo 3 round 25 clone 1 | This study | phML-35 | Derivative of phML-31 |
| Strain, strain background (T4) | T4 evo 4 round 25 clone 1 | This study | phML-36 | Derivative of phML-31 |
| Strain, strain background (T4) | T4 evo 5 round 25 clone 3 | This study | phML-37 | Derivative of phML-31 |
| Strain, strain background (T4) | T4 *tifA-1* | This study | phML-41 | N/A |

*Appendix 1 Continued on next page*

*Appendix 1 Continued*

| Reagent type (species) or resource | Designation | Source or reference | Identifiers | Additional information |
|---|---|---|---|---|
| Strain, strain background (T4) | T4 *tifA-2* | This study | phML-42 | N/A |
| Strain, strain background (T2) | T2 | ATCC | Cat# 11303-B2; phML-38 | N/A |
| Strain, strain background (T6) | T6 | ATCC | Cat# 11303-B6; phML-40 | N/A |
| Strain, strain background (RB69) | RB69 | Félix d'Hérelle Reference Center for Bacterial Viruses, Université Laval | HER# 158; phML-39 | N/A |
| Antibody | Anti-FLAG M2 magnetic beads (mouse monoclonal) | Sigma | Cat# M8823; RRID:AB_2637089 | Used 20 mL beads per 40 mL cells |
| Antibody | Recombinant anti-6X His tag rabbit antibody (rabbit monoclonal) | Abcam | Cat# AB200537 | Used at 1:5000× concentration |
| Antibody | DYKDDDDK tag rabbit mAb (rabbit monoclonal) | Cell Signaling Technology | Cat# 14793; RRID:AB_2572291 | Used at 1:5000× concentration |
| Antibody | Goat anti-rabbit IgG (H+L) secondary antibody, HRP (goat polyclonal) | Thermo Fisher Scientific | Cat# 32460; RRID:AB_1185567 | Used at 1:5000× concentration |
| Recombinant DNA reagent | pBAD33-*toxN* (plasmid) | Guegler and Laub, 2021 | ML-3343 | N/A |
| Recombinant DNA reagent | pEXT20-*toxI* (plasmid) | Guegler and Laub, 2021 | ML-3345 | N/A |
| Recombinant DNA reagent | pEXT20 (plasmid) | E. coli Genetic Stock Center | Cat# 12325; ML-1978 | N/A |
| Recombinant DNA reagent | pBR322 empty vector (plasmid) | Guegler and Laub, 2021 | ML-3348 | Derivative of pBR322 with pTet removed |
| Recombinant DNA reagent | pBR322-*toxIN* (plasmid) | Guegler and Laub, 2021 | ML-3346 | Derivative of ML-3348 |
| Recombinant DNA reagent | pBR322-*toxI-toxN*-His$_6$ (plasmid) | Guegler and Laub, 2021 | ML-3349 | Derivative of ML-3346 |
| Recombinant DNA reagent | pKVS45-*dmd*$_{T4}$ (plasmid) | This study | ML-3809 | Gene amplified from phML-31 |
| Recombinant DNA reagent (plasmid) | pKVS45-*tifA*$_{T4}$ | This study | ML-3810 | Gene amplified from phML-31 |
| Recombinant DNA reagent | pKVS45-*tifA*$_{T2}$ (plasmid) | This study | ML-3811 | Gene amplified from phML-38 |
| Recombinant DNA reagent | pKVS45-*tifA*$_{T6}$ (plasmid) | This study | ML-3812 | Gene amplified from phML-40 |
| Recombinant DNA reagent | pKVS45-*tifA*$_{RB69}$ (plasmid) | This study | ML-3813 | Gene amplified from phML-39 |
| Recombinant DNA reagent | pEXT20-*tifA*$_{T2}$ (plasmid) | This study | ML-3814 | Gene amplified from phML-38 |
| Recombinant DNA reagent | pEXT20-*tifA*$_{T4}$ (plasmid) | This study | ML-3815 | Gene amplified from phML-31 |
| Recombinant DNA reagent | pEXT20-*tifA*$_{T6}$ (plasmid) | This study | ML-3816 | Gene amplified from phML-40 |
| Recombinant DNA reagent | pEXT20-*tifA*$_{RB69}$ (plasmid) | This study | ML-3817 | Gene amplified from phML-39 |
| Recombinant DNA reagent | pKVS45-*tifA*$_{T4}$ DATG (plasmid) | This study | ML-3818 | Derivative of ML-3810 |
| Recombinant DNA reagent | pKVS45-*tifA*$_{T4}$ recoded (plasmid) | This study | ML-3819 | Derivative of ML-3810 |
| Recombinant DNA reagent | pKVS45-*tifA*$_{T4}$ ablated ToxN-site (plasmid) | This study | ML-3820 | Derivative of ML-3810 |

*Appendix 1 Continued on next page*

| Reagent type (species) or resource | Designation | Source or reference | Identifiers | Additional information |
|---|---|---|---|---|
| Recombinant DNA reagent | pKVS45-*tifA*$_{T4}$-FLAG (plasmid) | This study | ML-3821 | Derivative of ML-3810 |

*Appendix 1 Continued*

| Reagent type (species) or resource | Designation | Source or reference | Identifiers | Additional information |
|---|---|---|---|---|
| Recombinant DNA reagent | pBR322-$rIIA_{T4}$ (plasmid) | This study | ML-3822 | Gene amplified from phML-31 |
| Recombinant DNA reagent | pKVS45-$rIIB_{T4}$ (plasmid) | This study | ML-3823 | Gene amplified from phML-31 |
| Recombinant DNA reagent | pEXT20-$ipIII_{T4}$ (plasmid) | This study | ML-3824 | Gene amplified from phML-31 |
| Recombinant DNA reagent | pEXT20-$ipIII_{T4}DCTS$ (plasmid) | This study | ML-3825 | Derivative of ML-3824 |
| Recombinant DNA reagent | pBR322-$Dsr1_{ECOR17}$ (plasmid) | This study | ML-3826 | Gene amplified from ML-3794 |
| Recombinant DNA reagent | pKVS45-$nrdC.5_{T4}$ (plasmid) | This study | ML-3827 | Gene amplified from phML-31 |
| Recombinant DNA reagent | pCas9-$tifA_{T4}$-cr4 (plasmid) | This study | ML-3828 | Cas9 with guide targeting $tifA_{T4}$ |
| Commercial assay or kit | SuperSignal West Femto Maximum Sensitivity Substrate | Thermo Fisher Scientific | Cat# 34095 | N/A |
| Commercial assay or kit | Chameleon Duo Pre-Stained Protein Ladder | LI-COR | Cat# 928-60000 | N/A |
| Software, algorithm | ImageJ (v1.48) | NIH | https://imagej.nih.gov/ij/ | N/A |
| Software, algorithm | Bowtie2 (v2.1.0) | *Langmead and Salzberg, 2012* | http://bowtie-bio.sourceforge.net/bowtie2/index.shtml | N/A |
| Software, algorithm | SAMtools (v0.1.19) | *Li et al., 2009* | http://samtools.sourceforge.net/ | N/A |
| Software, algorithm | NumPy (v1.13.1) | *Charles, 2022*, RRID:SCR_008633 | https://github.com/numpy/numpy | N/A |
| Software, algorithm | Biopython (v1.65) | *Cock, 2022*, RRID:SCR_007173 | https://github.com/biopython/biopython | N/A |
| Software, algorithm | SciPy (v0.18.1) | *Reddy, 2022*, RRID:SCR_008058 | https://github.com/scipy/scipy | N/A |
| Software, algorithm | Jupyter Notebook | *Perez, 2022*, RRID:SCR_018315 | https://github.com/jupyter | N/A |
| Software, algorithm | *jackhmmer* | RRID:SCR_005305 | https://www.ebi.ac.uk/Tools/hmmer/search/jackhmmer | N/A |
| Software, algorithm | MUSCLE | RRID:SCR_011812 | https://www.ebi.ac.uk/Tools/msa/muscle/ | N/A |
| Software, algorithm | FastTree | RRID:SCR_015501 | http://www.microbesonline.org/fasttree/ | N/A |
| Software, algorithm | Cutadapt (v4.1) | RRID:SCR_011841 | https://pypi.org/project/cutadapt/ | N/A |
| Software, algorithm | BWA | *Li, 2022* | https://github.com/lh3/bwa | N/A |
| Software, algorithm | BCFtools | *Danecek, 2022*, RRID:SCR_005227 | https://github.com/samtools/bcftools | N/A |
| Software, algorithm | Geneious Prime 2021.2.2 | Dotmatics | N/A | N/A |
| Software, algorithm | Custom Python scripts | N/A | https://github.com/sriramsrikant/2022_T4-toxIN-evo; *Srikant, 2022* | N/A |
| Other | Agencourt AMPure XP | Beckman Coulter | A63880 | Magnetic beads used for sequencing library preparation |
| Other | Synergy H1 Hybrid Multi-Mode Microplate Reader | BioTek | N/A | Plate reader used for growth curves in *Figures 2C and 3C*, and *Figure 3—figure supplement 2B* |
| Other | NextSeq 500 Sequencing System | Illumina | Cat# SY-415-1001 | Instrument at MIT BioMicro Center used for DNA sequencing |
| Other | MiSeq Sequencing System | Illumina | Cat# SY-410-1003 | Instrument at MIT BioMicro Center used for DNA sequencing |
| Other | Bioruptor Plus | Diagenode | Cat# B01020001 | Sonicator used to shear phage gDNA prior to Illumina sequencing library preparation |
| Other | DTS-4 Digital Thermo Shaker | ELMI | N/A | 96-Well plate shaking incubator used for high-throughput phage evolution experiments |

