## [Editor Report]

This manuscript will be of interest to researchers in the phage-microbial host interaction field. Notably, the interplay between bacteria and their viral predators has regained broad interest in recent years given the discovery of numerous innate immunity-like phage defense systems. The identification of phage-mediated counter-defense strategies is therefore not only of prime importance for our basic understanding of predator-prey arms races but also for medical applications such as phage therapy.

---

## [Decision Letter]

**Decision letter after peer review:**

Thank you for submitting your article "The evolution of a counter-defense mechanism in a virus constrains its host range" for consideration by *eLife*. Your article has been reviewed by 3 peer reviewers, including Melanie Blokesch as Reviewing Editor and Reviewer #1, and the evaluation has been overseen by Gisela Storz as the Senior Editor.

The reviewers have discussed their reviews with one another, and the Reviewing Editor has drafted this to help you prepare a revised submission. Please note that no additional experiments will be required (the reviewer group agreed that the short reads combined with the distinctive PCR amplification pattern and Sanger sequencing are sufficient to support the claim of this study and that the suggested experiment to further increase ToxN abundance would be a nice add-on but isn't strictly required).

Essential revisions:

1) Figures need revisions for clarity (missing labels, low resolution, etc).

2) The authors should carefully update their Materials and methods section.

3) A few points need better discussion.

4) The accessibility of raw data on github, as stated in the manuscript, should be double-checked.

*Reviewer #2 (Recommendations for the authors):*

The manuscript is well written and the storyline is easy to follow. We also appreciate the schemes to explain more difficult points (e.g. Figure S1D and Figure S4D).

We find it remarkable that the acquisition of resistance in the toxin-expressing host was acquired in 5 evolved clones through segmental amplification of the dmd-tifA locus (versus mutations in the promoter sequence to increase transcription levels for example). This illustrates the extent of recombination within phage genomes thanks to regions of microhomology, in comparison to the pervasiveness of point mutations in bacterial genomes.

Interestingly, tifA is surrounded by further genes encoding anti-TA systems suggesting the clustering in anti-TA islands – an observation that could guide further investigations.

Overall, this is without question an interesting example, but it describes a very specific phenomenon, namely the adaptation of phage T4 to an *E. coli* strain expressing a plasmid-based toxN. Their results also show that native tifA expression is sufficient to overcome ToxN production from the chromosome.

1. It is puzzling that the control evolved clone (without toxin) showed no resistance clones appearing, even after 25 rounds of infection (figure S4A). Is the emergence of resistance so rare in T4?

2. It would be maybe advisable to be precise that RnlAB is a type II TA system. This was not clear in the manuscript.

3. The relative amounts of ToxN and TifA seem to be a strong determinant of the success of infection. Would the evolved clones still show infection of an *E. coli* expressing toxN on higher copies?

l. 27-38, insights in bacterial immune systems are currently exploding. Therefore, I would suggest citing some even more recent literature (e.g. doi: 10.1038/s41577-022-00705-4, doi: 10.1016/j.cell.2021.12.029)

*Reviewer #3 (Recommendations for the authors):*

The authors show plenty of agarose gel electrophoresis pictures. However, the ladder and the size of the expected PCR products are not annotated.

The data points of the growth curves should be ba accessible at github. However, I cannot find them. The authors should add clear titles to the datasets to improve data transparency.

Figure S3A has a poor resolution which hampers readability.

---

## [Author Response]

Reviewer #2 (Recommendations for the authors):The manuscript is well written and the storyline is easy to follow. We also appreciate the schemes to explain more difficult points (e.g. Figure S1D and Figure S4D).We find it remarkable that the acquisition of resistance in the toxin-expressing host was acquired in 5 evolved clones through segmental amplification of the dmd-tifA locus (versus mutations in the promoter sequence to increase transcription levels for example). This illustrates the extent of recombination within phage genomes thanks to regions of microhomology, in comparison to the pervasiveness of point mutations in bacterial genomes.Interestingly, tifA is surrounded by further genes encoding anti-TA systems suggesting the clustering in anti-TA islands – an observation that could guide further investigations.Overall, this is without question an interesting example, but it describes a very specific phenomenon, namely the adaptation of phage T4 to an *E. coli* strain expressing a plasmid-based toxN. Their results also show that native tifA expression is sufficient to overcome ToxN production from the chromosome.1. It is puzzling that the control evolved clone (without toxin) showed no resistance clones appearing, even after 25 rounds of infection (figure S4A). Is the emergence of resistance so rare in T4?

We were never able to observe *toxIN* resistance by T4 in the absence of our evolution experiment, suggesting that without serial passaging these mutations are exceptionally rare.

2. It would be maybe advisable to be precise that RnlAB is a type II TA system. This was not clear in the manuscript.

We have adjusted the text in the introduction to clarify this point.

3. The relative amounts of ToxN and TifA seem to be a strong determinant of the success of infection. Would the evolved clones still show infection of an *E. coli* expressing toxN on higher copies?

This is an interesting idea and seems likely given the results shown here though we did not formally test it.

l. 27-38, insights in bacterial immune systems are currently exploding. Therefore, I would suggest citing some even more recent literature (e.g. doi: 10.1038/s41577-022-00705-4, doi: 10.1016/j.cell.2021.12.029)

These citations have been added to the introduction.

Reviewer #3 (Recommendations for the authors):The authors show plenty of agarose gel electrophoresis pictures. However, the ladder and the size of the expected PCR products are not annotated.

We have clarifying labels to the agarose gels throughout the figures to annotate relevant ladder positions. When possible, we have also added labels to PCR bands. Because each evolved T4 clone has differently-sized deleted and amplified regions, it is difficult to do this in some figures without making them hard to read. To help the reader better interpret the remaining agarose gels, we have added scale bars above our diagrams when the scale is not demonstrated elsewhere (i.e., in neighboring read coverage diagrams).

The data points of the growth curves should be ba accessible at github. However, I cannot find them. The authors should add clear titles to the datasets to improve data transparency.

Raw data for the growth curves in Figures 2C, 3C, and S3C are indeed accessible on Github, in a document called ‘220426_CKG_growthcurve_rawdata.xlsx’ in the sub-directory data/plot_data. Upon crafting our reviewer responses, we verified that this document is accessible and that the data are clearly labeled. We apologize if the reviewer was unable to find them. To make titles clearer, we have added a table of contents document to the Github repository to detail the type of data and the figure that each dataset corresponds to in the manuscript.

Figure S3A has a poor resolution which hampers readability.

We have uploaded a higher-resolution version of this figure.